# High-resolution genome topology of human retina uncovers super enhancer-promoter interactions at tissue-specific and multi-factorial disease loci

Claire Marchal[1,2,3], Nivedita Singh[1,3], Zachary Batz [1,3], Jayshree Advani[1,3], Catherine Jaeger [1], Ximena Corso-Díaz[1] & Anand Swaroop [1] ✉

Chromatin organization and enhancer-promoter contacts establish unique spatiotemporal gene expression patterns in distinct cell types. Non-coding genetic variants can influence cellular phenotypes by modifying higher-order transcriptional hubs and consequently gene expression. To elucidate genomic regulation in human retina, we mapped chromatin contacts at high resolution and integrated with super-enhancers (SEs), histone marks, binding of CTCF and select transcription factors. We show that topologically associated domains (TADs) with central SEs exhibit stronger insulation and augmented contact with retinal genes relative to TADs with edge SEs. Merging genome-wide expression quantitative trait loci (eQTLs) with topology map reveals physical links between 100 eQTLs and corresponding eGenes associated with retinal neurodegeneration. Additionally, we uncover candidate genes for susceptibility variants linked to age-related macular degeneration and glaucoma. Our study of high-resolution genomic architecture of human retina provides insights into genetic control of tissue-specific functions, suggests paradigms for missing heritability, and enables the dissection of common blinding disease phenotypes.

The three-dimensional (3D) architecture of the human genome is regulated across multiple levels of organization, yielding precise spatiotemporal patterns of gene expression for morphogenesis and functional specification[1,2]. Gene regulation occurs within transcriptional units through productive enhancer-promoter contacts and/or by inclusion within transcriptionally active membraneless structures, called phase-separated condensates[3–6]. The transcriptional units are contained within megabase-sized self-interacting chromatin structures known as topologically associated domains (TADs). The boundaries of TADs are enriched for binding of structural proteins including CTCF and cohesin[7–9]. Genome organization also exhibits A (active) and B

(inactive) chromatin compartments, which display distinct patterns of DNA replication and transcription along with differences in regulatory marks[10–13]. Across the 3D hierarchy, genome topology undergoes dynamic and contextual physical alterations in distinct tissues and cell types[14,15]. These adaptations correlate with activation of specific *cis*-regulatory elements (CREs) that contribute to the establishment of unique gene expression patterns[14,16,17]. Cell-type specific gene expression is further orchestrated by super-enhancers (SEs), regulatory regions spanning over tens of kilobases that are highly enriched for master transcription factor (TF) binding and co-localized with the active histone mark H3K27Ac[18–20].

[1]Neurobiology, Neurodegeneration and Repair Laboratory, National Eye Institute, National Institutes of Health, MSC0610, 6 Center Drive, Bethesda, MD 20892, USA. [2]In silichrom Ltd, First Floor, Angel Court, 81 St Clements St, Oxford OX4 1AW, UK. [3]These authors contributed equally: Claire Marchal, Nivedita Singh, Zachary Batz, Jayshree Advani. ✉e-mail: swaroopa@nei.nih.gov

Over the past decade, millions of human genetic variations have been cataloged[21,22], permitting exploration of evolutionary divergence as well as healthy and disease phenotypes. Genome-wide association studies (GWAS) of common multifactorial retinal diseases afflicting adult human populations, such as age-related macular degeneration (AMD) and glaucoma, have predominantly identified common variants in non-coding regions of the genome[23,24]. However, the biological relevance of association signals has been difficult to assess because of the local linkage disequilibrium (LD) in the region of lead variants, hindering identification of specific causal genes and variants[25]. Even in Mendelian retinal diseases with over 200 associated genes identified (RetNet; https://sph.uth.edu/retnet/disease.htm), causal mutations can only be identified in about half of the patients (primarily in European population)[26] and genotype-phenotype correlations have been difficult to decipher. Non-coding variants in *cis*-regulatory regions may lead to variable penetrance of pathogenic coding mutations in their target genes[27], genetic epistasis[28], or account for missing heritability[29]. Widespread variability in gene expression observed in humans can be assigned to *cis*- or *trans*-acting variants (expression quantitative trait loci, eQTLs) and epistasis[30]. Elucidating how trait-associated genetic variations impact the regulatory landscape and consequently phenotypes requires understanding the 3D genome topology in relevant tissues and cell types.

Here, we report a high-resolution chromatin contact map of the adult human retina by performing Hi-C[1]. Further integration of chromatin contacts with histone marks, chromatin accessibility, selected TF binding, and gene expression datasets reveals targets of CREs and uncovers properties of 3D chromatin organization of SEs in human retina. Finally, we combine the resulting retinal genomic regulation network with eQTLs and genetic variants identified through GWAS of AMD and glaucoma. Thus, our analysis of regulatory 3D genome architecture contributes to better understanding of genetic control of human retinal phenotypes.

## Results

### Deep Hi-C sequencing identifies chromatin structures in human retina at 5 kb resolution

Figure 1A illustrates the design of our study to elucidate regulatory networks controlling gene expression in the human retina and their potential disruption by genetic variants associated with clinical phenotypes. Briefly, we have generated a high-resolution chromatin contact map of the human retina using Hi-C and integrated this data with histone marks, chromatin accessibility, CTCF, and binding of selected TFs. We then explored the genome topology of eQTLs and variants associated with AMD and glaucoma (Fig. 1A).

To decipher the 3D organization of chromatin, we performed Hi-C analysis of four independent postmortem human retina samples. The Hi-C data exhibited high similarity among samples with a stratified correlation coefficient (SCC) of >0.97 for all autosomes. We therefore combined the data from all samples to obtain a total sequencing depth of 1.148 billion read pairs. A high proportion of valid interactions (>95% of the total mapped interactions) and typical percentage of trans-interactions (23.6%; Supplementary table 1) indicated the high quality of our dataset. We obtained 704 million valid chromatin contacts (Supplementary table 1), which are equally distributed among the chromosomes, attaining a resolution of 3 kb (Fig. 1B, C, Supplementary fig. 1A). Our data identified 67,841 significant chromatin contacts and 2948 TADs (Fig. 1A, C, D).

As predicted, the genes expressed in the retina are detected in the A chromatin compartment, while most silent genes appear in the B compartment (Fig. 1C; Supplementary Fig. 1B, C). For example, *NRL* is expressed in rods, *ESRRB* in rods and horizontal cells, and *VSX2* in bipolar and Muller glia cells; each of the three genes is present within a well-defined A compartment TAD and participates in extensive intra-TAD chromatin looping (Fig. 1D), which is likely required for

qualitatively and quantitatively precise regulation[6]. We then characterized the loops associated with expressed chromatin at the genome-wide level. Intra-A compartment loops are shorter compared to the intra-B (Tukey HSD: $p < 0.001$; 95% CI 416–449 kb) and the A-B compartment loops (Tukey HSD: $p < 0.001$; 95% CI 308–398 kb; Fig. 1E). These intra-A loops, corresponding to the active chromatin, span less than 1 Mb on average (Fig. 1E) and are mostly constrained within TADs (Supplementary fig. 1D).

### Human retina chromatin topology is tissue-specific and conserved in mouse

To assess whether the identified chromatin features are specific to the retina, we compared our dataset with Hi-C datasets from human neurons (anterior cingulate cortex (ACC) neurons)[31], GM12878 lymphoblastoid immortalized cells (LCL)[2], and colon cancer cells (HCT116; generated in this study). The correlation, as measured by SCC, is the highest among our four retina samples (Fig. 2A). Retina compartments reveal high correlation with those of ACC neurons though limited to few chromosomes, likely because of the improper calling of compartments in the low-resolution ACC dataset (Fig. 2B, Supplementary fig. 2A). The compartments in the non-transformed LCL dataset are broadly similar with retinal compartments, whereas the cancer cells exhibit the lowest correlation (Fig. 2B).

We hypothesized that regions in the A compartment, which are detected in retina but not in other tissues, are related to retina-specific functions. Indeed, retina-specific genes such as *OTX2*, *EYS*, *PDC* are present in the A compartment only in the retina (Fig. 2C). Given the high correlation between retina and ACC neurons, we compared the contact maps of *PAX6*, *OTX2* and *CRX* between the two datasets. As predicted, interactions at the *PAX6* locus are similar in neurons and retina concordant with its expression in both tissues (Fig. 2D, left panel), whereas loci encoding *OTX2* and *CRX*, two key retinal TFs, demonstrate a high number of local interactions in the retina but not in neurons (Fig. 2D, central and right panels).

Distinct chromatin interactions at retinal genes suggest their important role in tissue-specific gene regulation. We then assessed whether these interactions are conserved between human and mouse retina. To quantify shared chromatin interactions, we evaluated the number of gene pairs present in the same TAD in our human and a previously reported mouse retina Hi-C datasets[32]. We observed that 21.8% of mouse gene pairs in a mouse TAD are also localized in the corresponding human TAD (Fig. 2E top panel), and that 35.7% of mouse gene pairs interacting through a chromatin loop in the retina are also interacting in human (Fig. 2E bottom panel). These numbers likely represent an underestimate of shared chromatin structure due to the relatively lower resolution of the mouse retina Hi-C data. As an example, the *PITX2/EGF* locus, located in a human/mouse syntenic region, extends over 3 TADs in the human retina. This region encompasses several expressed genes, with *EGF* and *ENPEP* genes interacting together, each at a TAD boundary (Fig. 2F). Self-interacting domains corresponding to the human TADs (Fig. 2F, dashed lines) are detectable in the mouse retina (Fig. 2F), even though not identified as TADs in the original study. Furthermore, in both species, chromatin loops are observed between *EGF* and *ENPEP* (Fig. 2F). These findings show that retina-expressed genes exhibit a conserved chromatin topology, underlining its importance for the regulation of retinal genes and likely involving tissue-specific CREs.

### Retinal SEs overlap with highly expressed tissue-specific genes and are enriched for accessible retinal TF binding motifs

Chromatin looping plays a crucial role in promoting physical interactions between CREs and their target genes to precisely control gene expression. To identify retinal CREs, we determined chromatin states using a Hidden Markov Model (ChromHMM)[33] based on chromatin accessibility (Supplementary fig. 3A) and active (H3K4me3, H3K27Ac,

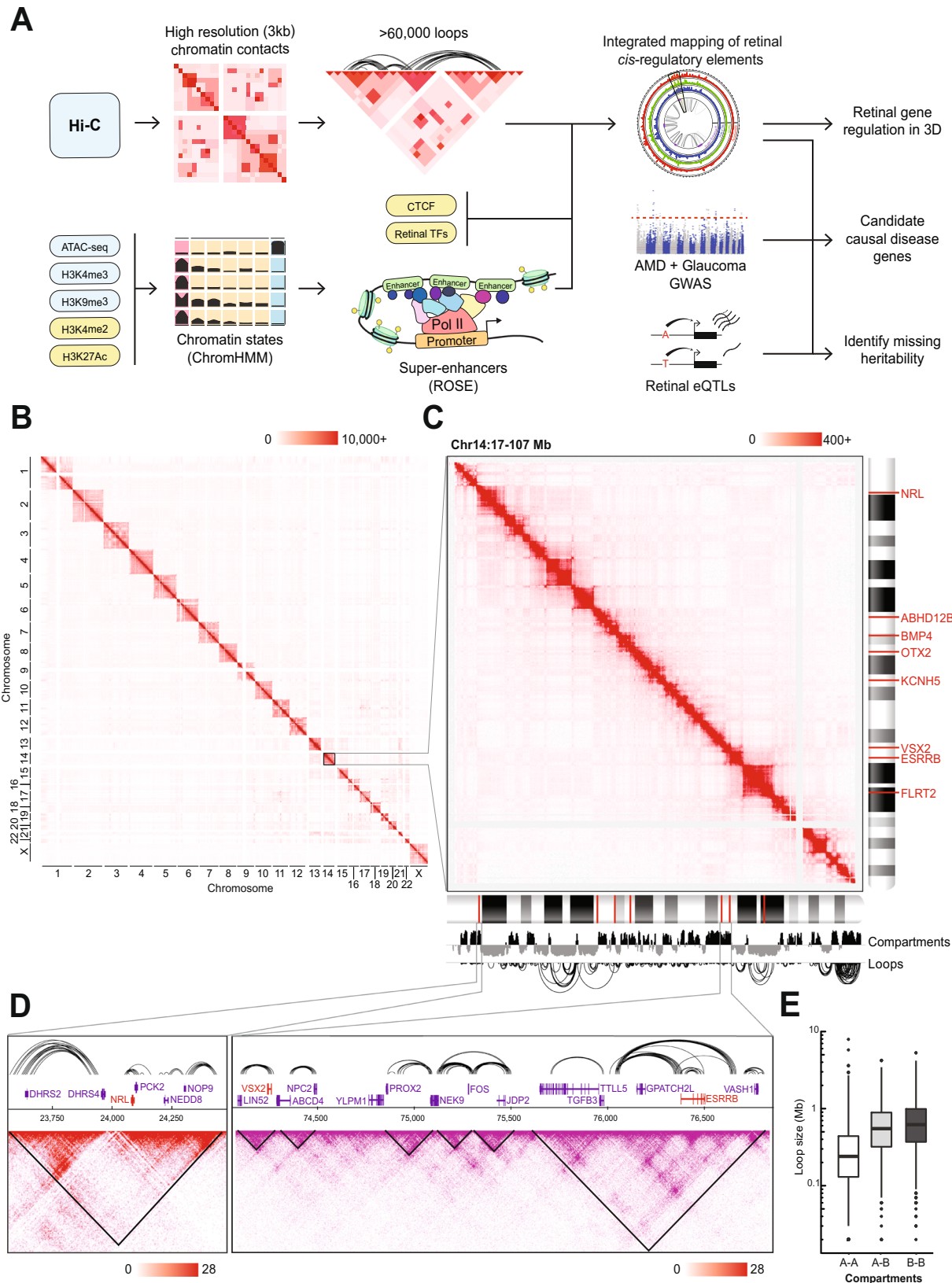

H3K4me2[34]) and repressive (H3K9me3) histone marks. We set the model for 10 chromatin states (see Methods, and Supplementary fig. 3B) and performed manual annotation. Of these states, we removed three with low signal and annotated one as promoters (high chromatin accessibility and enriched for all active marks), five as active or poised enhancers (high chromatin accessibility and enriched for all

active marks except H3K4me3), and one as heterochromatin (enriched for H3K9me3) (Fig. 3A). Promoters and active or poised enhancers are enriched for binding of CTCF and key retinal TFs such as CRX, NRL, OTX2, MEF2D, CREB, and RORB (Supplementary fig. 3C). Moreover, active and poised enhancers are enriched at regions upstream to TSS, whereas promoters are enriched at TSS only (Supplementary fig. 3D).

**Fig. 1 | High-resolution Hi-C identifies human retinal chromatin structures.**
**A** Schematic of project workflow. Input data on blue backgrounds were generated in this manuscript; input data on yellow backgrounds were downloaded from public databases. Hi-C contact maps of observed contacts across (**B**) all chromosomes, (**C**) the q arm of chromosome 14, and (**D**) the *NRL* and *VSX2/ESRRB* loci. Scale represents the number of raw contacts. Loops at the *NRL* and *VSX2/ESRRB* loci are plotted on the top; contact maps are below with TADs plotted as black triangles.

**E** Distribution of chromatin loop sizes within and across A/B compartments (n = 15,441 loops within A compartment, n = 4318 loops within B compartments, n = 2697 loops in A-B compartments). Boxplots represent the median and inter-quartile range (IQR); whiskers mark 1.5x the IQR; data beyond 1.5x the IQR are plotted as individual points. The three groups differ significantly (one-way ANOVA, $F_{2,22453}$ = 2232, $p < 0.001$). Abbreviations: ROSE Rank Ordering of Super-Enhancers, AMD Age-related macular degeneration, GWAS Genome-wide association study.

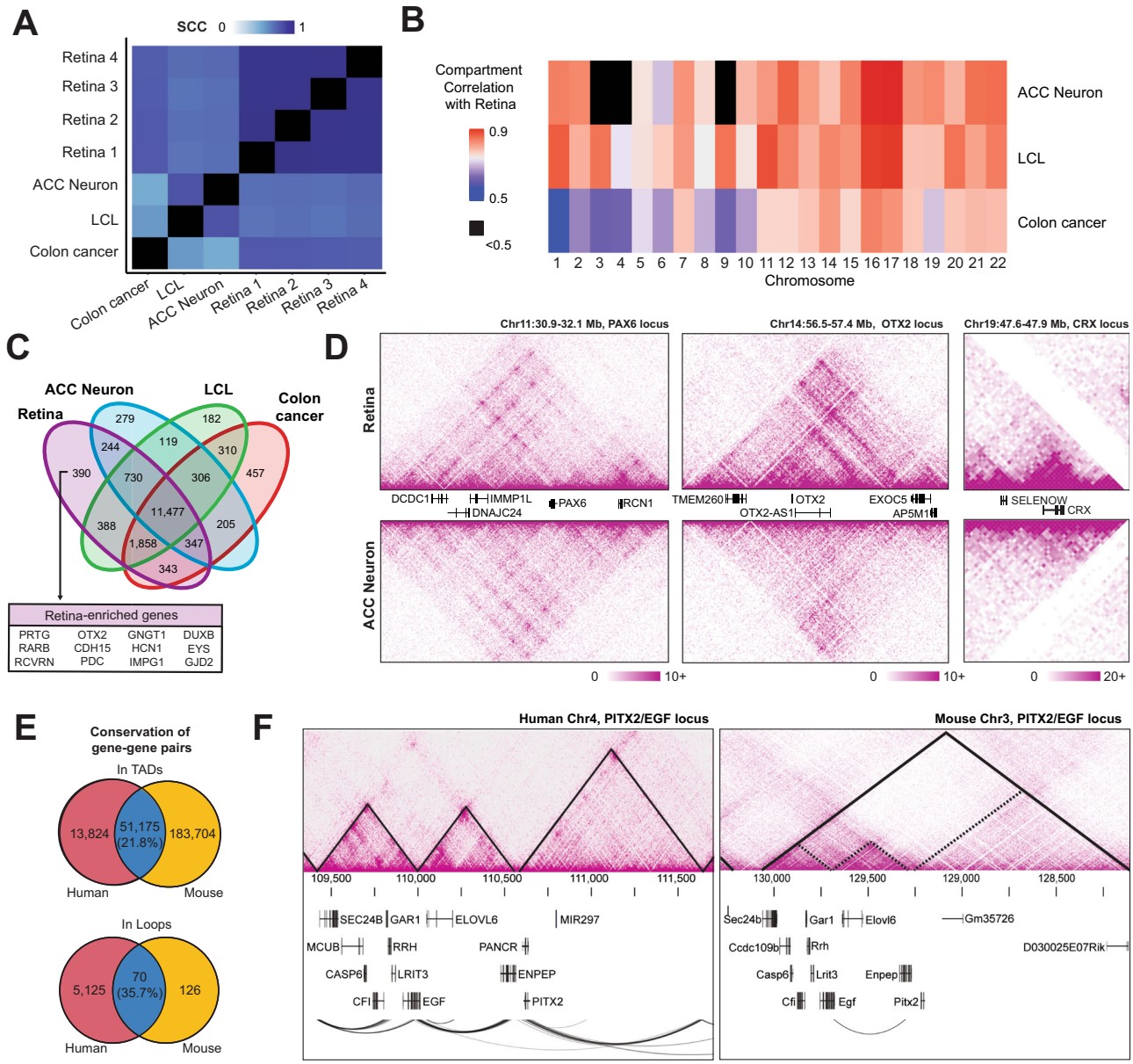

**Fig. 2 | Retinal chromatin interactions are tissue specific and conserved in mouse. A** Stratum-adjusted correlation coefficient (SCC) showing similarity of contact maps between tissue types averaged across all autosomal chromosomes. **B** Chromosome-level correlation of A/B compartments between retina and various tissues. **C** Venn diagram of gene transcription start sites present in the A compartment across sample types. A selection of retina-enriched genes (see methods) exclusively identified as A compartment in retina samples are highlighted. **D** Comparison of Hi-C contact maps for retina and ACC neuron at the *PAX6*, *OTX2* and *CRX* loci. Scales represent the number of Knight-Ruiz (KR) normalized contact

counts. **E** Human-mouse conservation of gene-gene shared occupancy within a TAD (top) and gene-gene contacts via Hi-C loop calls (bottom). Percentages given are relative to the number of mouse gene-gene pairs. **F** Hi-C contact maps and loops in a syntenic region of the human and mouse genomes. Solid lines on contact maps indicate computationally called TADs; dashed lines on the mouse contact map show human TAD boundaries overlaid on mouse genome. Scales represent the number of KR normalized contacts. Abbreviations LCL Lymphoblastoid B-cell line, ACC Anterior cingulate cortex, TADs Topologically associating domains.

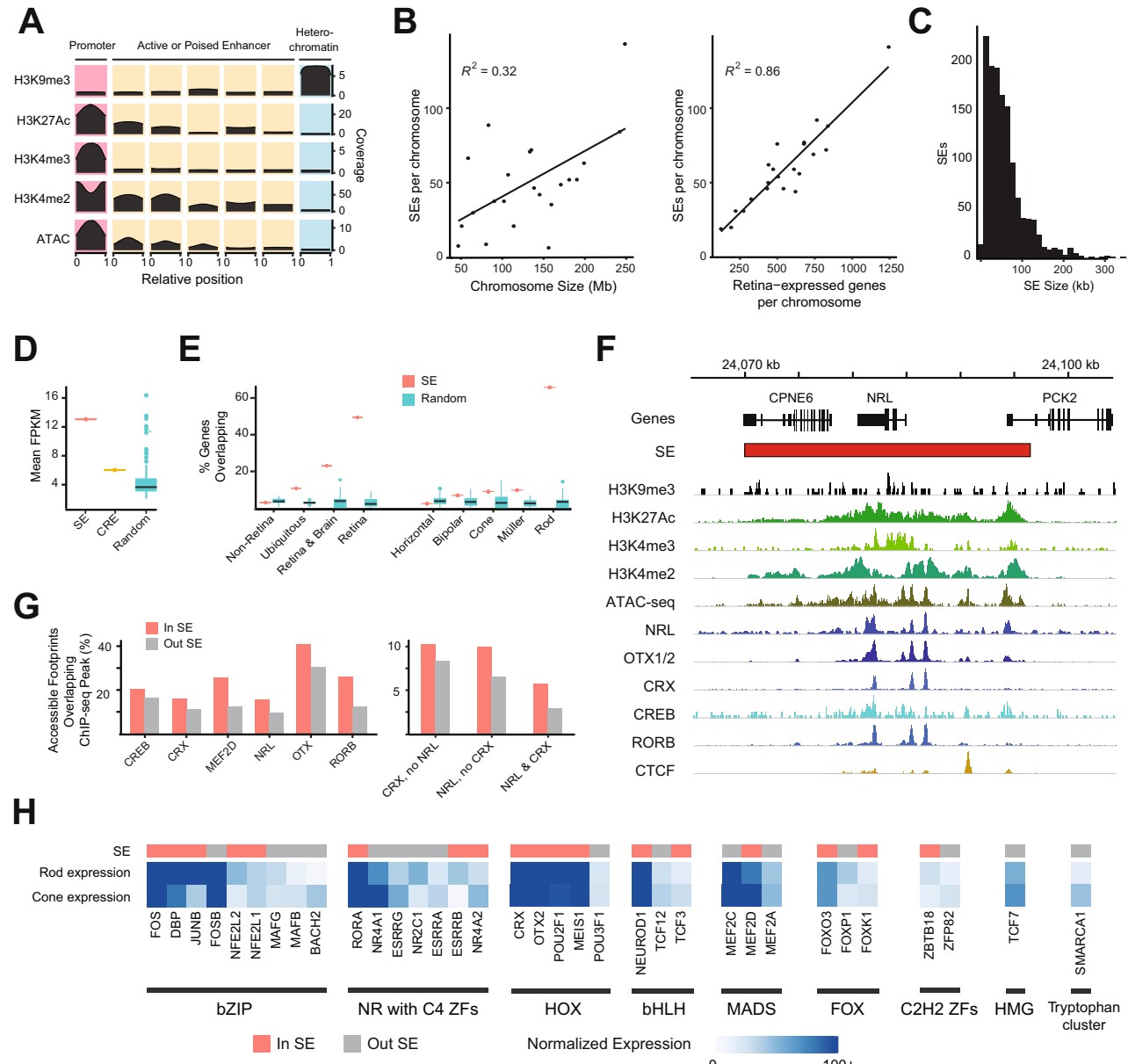

**Fig. 3 | Retinal SE identification and characterization. A** Chromatin states identified from histone marks and ATAC-seq data (low signal states not shown, see Fig. S3). Plots in each cell indicate the mean coverage for each mark across the genomic regions aligned and resized to a 0 to 1 scale, e.g., regions assigned the heterochromatin state contain uniformly high H3K9me3 coverage across the entire region and low coverage of all other marks. **B** Number of SE per chromosome versus chromosome size (left) and versus number of retina-expressed genes (right). Correlation is measured as Pearson's $r^2$. **C** Size distribution of SEs in retina. **D** Mean expression of genes with transcription start sites (TSS) in SE ($n = 1$), CRE ($n = 1$), and random SE-sized genomic regions ($n = 100$). Boxplots represent the median and interquartile range (IQR); whiskers mark 1.5x the IQR; data beyond 1.5x the IQR are plotted as individual points. **E** Percentage of genes in various enrichment groups (see methods) with at least one TSS located in a SE ($n = 1$) or random SE-sized region ($n = 100$). Boxplots represent the median and interquartile range (IQR); whiskers mark 1.5x the IQR; data beyond 1.5x the IQR are plotted as individual points. **[F]** SEs, histone marks, chromatin accessibility, and TF residency for the *NRL* locus.

**G** Percentage of accessible genomic footprints (defined via ATAC-seq) inside and outside of SEs which contain ChIP-seq narrow peaks for the indicated TF(s). **H** Selection of TF motifs enriched in SE-overlapping accessible genomic footprints containing NRL and CRX relative to accessible footprints without NRL or CRX. The bar on top indicates whether the gene coding for the TF is overlapping a SE (red) or not (grey). The heatmap shows normalized gene expression in rods and cones from the Human Protein Atlas. TF families as defined by TFClass are indicated along the bottom edge. Abbreviations ATAC-seq Assay for transposase-accessible chromatin sequencing, SE Super-enhancer, FPKM Fragments per kilobase million, TSS Transcription start site, ChIP-seq Chromatin immunoprecipitation sequencing, C2H2 ZFs C2H2 zinc finger factors, bZIP Basic leucine zipper factors, NR with C4 ZFs Nuclear receptors with C4 zinc fingers, HOX Homeodomain factors, bHLH Basic helix-loop-helix factors, MADS MADS-box factors, HMG High-mobility group domain factors, FOX Fork head/winged helix factors, RHR *Rel* homology region factors.

We then identified 1,325 SEs using the density of H3K27Ac at the retinal CREs (promoters and active or poised enhancers) after removing loci corresponding to TSS. The distribution of SEs along chromosomes is not homogenous and shows a very low correlation ($R^2 = 0.31$) between the number of SEs per chromosome and the chromosome size (Fig. 3B left panel and Supplementary fig. 3E). Interestingly, however, we detected a high correlation ($R^2 = 0.86$) between the number of retina-expressed genes and the number of SEs per chromosome (Fig. 3B, right panel). Retinal SEs also form large chromatin domains spanning over 10 to 300 kb (Fig. 3C).

Integration of SEs with transcriptome data revealed 2.17x higher average expression of genes overlapping with SEs compared to those with CREs and 2.89x higher than 100 sets of random SE-sized loci (henceforth called "random regions", see Methods for details) (Fig. 3D, Supplementary Data 1). Notably, nearly 50% of retina-enriched genes and 70% of rod-enriched genes show an overlap with a SE compared to ~5% with random regions (Fig. 3E). The key photoreceptor-specific gene *NRL* is included within a large SE enriched for active histone marks and accessible chromatin regions (Fig. 3F). Accessible footprints within SEs are enriched for binding peaks of retinal TFs such as NRL, CRX, OTX1/2, CREB and RORB (Fig. 3G). Furthermore, hotspots of TF binding sites are present predominantly in SEs compared to CREs or random regions (Supplementary fig. 3F). We also confirmed that accessible chromatin regions in retinal SEs are enriched for binding of two key photoreceptor TFs, CRX and NRL, alone or in combination (Fig. 3G, right panel). We discovered that CRX and NRL bound loci are enriched for binding motifs of other TFs involved in retinal gene regulation including OTX2, MEF2C, NEUROD1 and MAFG (Fig. 3H, Supplementary fig. 3G, Supplementary Data 2). In addition, we uncovered an enrichment of motifs for retina-expressed TFs, e.g., ZBTB18, for which a specific function has not been delineated in retinal cell types (Fig. 3H, Supplementary Data 2). The genes encoding many of these TFs with enriched binding motifs also overlap SEs, suggesting their potential role in maintaining retinal homeostasis (Fig. 3H, Supplementary Data 2).

### SEs are enriched for intra-TAD looping and contacts with regulatory elements

SEs may contact other regulatory regions to form large transcriptional units capable of regulating multiple genes, as previously observed in cell lines[35,36]. Integrated analysis revealed that retinal SEs are enriched for shorter chromatin loops compared to the random regions ($t_{99} = 69.46$; $p < 0.001$; Fig. 4A) and show extensive intra-SE looping (Fig. 4B). Distinctly, nearly all SE-interacting loops are <1 Mb (Fig. 4A) indicating that SEs primarily harbor interactions within TADs. To validate this observation, we quantified the percentage of intra-TAD loops overlapping with SEs. We detected an enrichment of loops interacting within the same TAD compared to the random regions ($t_{99} = −24.76$; $p < 0.001$), whereas we observed fewer than expected loops interacting across TADs ($t_{99} = −18.81$; $p < 0.001$; Fig. 4C).

To quantify the significance of chromatin looping at SEs, we defined SE interactions as those with another regulatory element or with a target gene. We discovered that SEs are enriched for regulatory interactions compared to random regions, making direct contact with 3,059 unique CREs ($t_{99} = −361$; $p < 0.001$), 1,701 unique TSS ($t_{99} = −346$; $p < 0.001$), and 217 unique SEs ($t_{99} = −651$; $p < 0.001$; Fig. 4D). Remarkably, the genes specifically expressed in the retina and/or brain are mostly overlapping with SEs, whereas the genes interacting but not overlapping with SEs are enriched for ubiquitous genes (Supplementary fig. 4A). Some SEs interact with many CREs and thereby likely play important roles in coordinating gene expression patterns. For example, the SE overlapping with *MIR9-2*, a non-coding gene crucial for neuronal development[37] localizes at a TAD boundary and contacts several CREs (Fig. 4E). We also identified several clusters of SEs interacting with one another via chromatin looping near retina-enriched genes, including *VEGFA* (Fig. 4F) and *UBE4B* (Fig. 4G).

### Intra-TAD positioning of SEs is associated with boundary insulation and biological function of target genes

To explore the chromatin landscape around SEs, we assessed physical properties of the corresponding host TADs. We first compared the TAD size in relation to the position of SE within the TAD. We observed that, within the A compartment, TADs containing SEs are larger than TADs without SEs, except when the SE is located at one of the TAD edges (Fig. 5A, left panel). We then calculated the insulation score,

which, for each genomic locus, quantifies the interaction frequency between neighboring loci; a lower score indicates stronger insulation. Our analysis showed that the presence of SEs at the edge of TADs is associated with weaker insulation, i.e., increased contact between neighboring TADs (Fig. 5A, right panel). We then evaluated whether SE positioning within TADs was associated with the number of inter-TAD contacts. We identified more chromatin interactions within SE-containing TADs compared to other TADs, despite no significant difference in loop size among A-compartment TADs (Fig. 5B). In concordance with the observed weaker insulation, TADs with a SE at one edge have a significantly higher proportion of loops crossing the TAD boundary compared to other TADs (Fig. 5B).

Next, we looked at the function of the SE target genes, i.e., genes with TSS overlapping or interacting with SE. We observed that TADs with edge SEs are enriched for stress response genes indicating the need for more dynamic and transitional interactions (Fig. 5C), whereas retinal genes are primarily enriched in TADs with central SEs (Fig. 5C, Supplementary fig. 5A). The SE overlapping the stress-response gene *FOS* is an example of an edge SE within a low insulation TAD that may be affected by variation in regions extending beyond their own TAD boundaries (Fig. 5D). Similarly, at the *ATF4* locus, a SE lying at the edge of a lowly insulated TAD is in contact with regions outside of the TAD (Supplementary fig. 5B).

### Integration of chromatin loops, SEs and CREs with retinal eQTLs

eQTLs link specific genetic variants to changes in expression of a target gene (henceforth called "eGene"). To identify eQTLs that are potentially relevant for retinal gene regulation, we incorporated 14,859 previously reported retina eQTLs[38] with our dataset and observed that 77% of these localize to the A compartment compared to 18.5% in the B compartment (Fig. 6A, Supplementary Data 3). Interestingly, the A compartment includes 82.9% of eQTLs linked to retinopathy genes (RetNet, https://sph.uth.edu/RetNet/), 90% of those associated with AMD loci[23], and 53.5% eQTLs correlated to glaucoma loci[24] (Fig. 6A). eQTLs are enriched for variant-eGene pairs sharing the same TAD (67.1% of all eQTLs, 75.4% of those in retinopathies, 73.3% AMD-associated eQTLs, and 79.2% glaucoma eQTLs) compared to randomly generated TADs (Fig. 6B; Supplementary Data 3). Thus, most retinal eQTLs are present within the active chromatin compartment and reside in the same TAD, providing a direct mechanism to explain the impact of a variant on the eGene.

We then assessed the physical proximity between a variant and the promoter region (±2.5 kb from TSS) of its eGene, as described[39]. This analysis allowed us to discriminate between promoter eQTLs, distal eQTLs, and distal eQTLs interacting with a promoter (pieQTLs, see methods) (Fig. 6C). We identified 2,374 eQTLs overlapping with a loop foot; of these, 1,275 are pieQTLs and 100 interact with their own eGene promoter (Fig. 6D). Among these pieQTLs, 27 are associated with retinopathies, including 3 that interact with their own eGene promoter (*MYO7A*, *MERTK* and *PRPH2* genes). Finally, 4 pieQTLs are each associated with AMD and glaucoma loci (Supplementary Data 3).

Additional analysis identified 2,410 and 880 eQTLs in CREs (CRE-eQTLs) and SEs (SE-eQTLs), respectively, with 58.5% of CRE-eQTLs and 69.3% of SE-eQTLs also intersecting the associated eGene (Fig. 6E, Supplementary Data 3). A majority of retinopathy-associated CRE-eQTLs (39/60) and SE-eQTLs (38/42) overlaps the respective eGene (Fig. 6E, Supplementary Data 3). Furthermore, we discovered 3 CRE-eQTLs and 2 SE-eQTLs in AMD-associated loci and 17 CRE-eQTLs and 5 SE-eQTLs in glaucoma-associated loci (Fig. 6E). In accordance with their potential role in gene regulation, all CRE- and SE-overlapping eQTLs exhibit prominent H3K27Ac marks, with promoter eQTLs showing even higher levels compared to distal eQTLs, though not statistically significant for the disease-associated eQTLs, due to the lower number of eQTLs (all eQTLs at CREs: $t_{1059.5} = −14.266$, $p < 2.2e\text{-}16$,

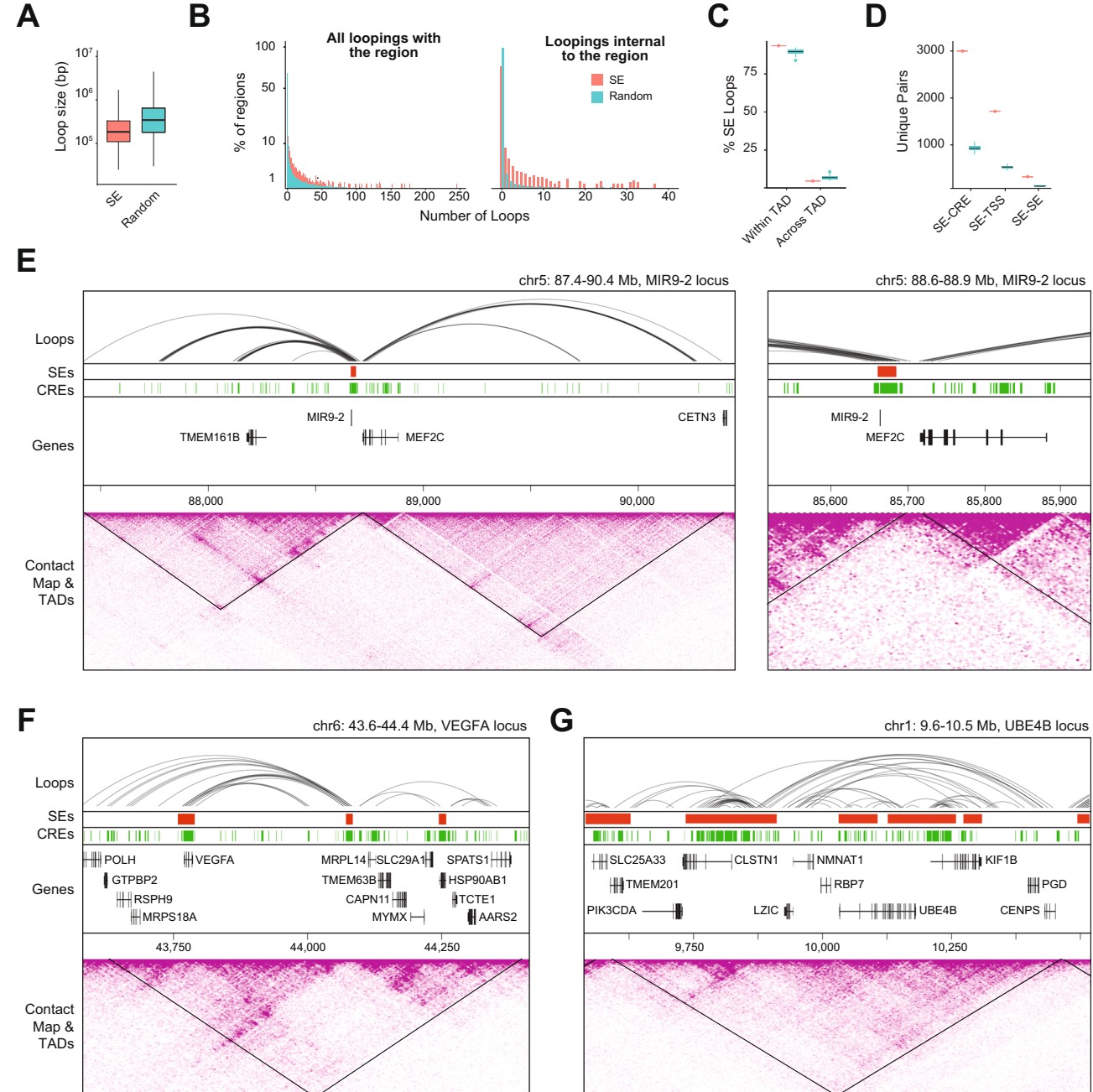

**Fig. 4 | SEs display a distinct chromatin looping pattern. A** Size distribution of loops intersecting with at least one SE ($n = 14{,}491$ loops) or random SE-sized genomic region ($n = 299{,}795$ total loops across 100 randomly generated regions, see methods). Boxplots represent the median and interquartile range (IQR); whiskers mark 1.5x the IQR; data beyond 1.5x the IQR are plotted as individual points. **B** Distribution of loops per SE or random SE-sized genomic region either making contact with at least one foot touching an SE/random region (left) or both feet contained within a single SE/random region (right). **C** Percentage of loops in contact with SEs ($n = 1$) or random regions ($n = 100$) that cross TAD boundaries. Boxplots represent the median and interquartile range (IQR); whiskers mark 1.5x the IQR; data beyond 1.5x the IQR are plotted as individual points. **D** Number of unique pairs of features connected by chromatin looping ($n = 1$ SEs dataset, $n = 100$ random regions datasets). Boxplots represent the median and interquartile range (IQR); whiskers mark 1.5x the IQR; data beyond 1.5x the IQR are plotted as individual points. **E–G** Chromatin loops, SEs, CREs, TADs, and Hi-C contact maps for the [**E**] *MIR9-2* (zoom-in on right panel), [**F**] *VEGFA*, and [**G**] *UBE4B* loci. Abbreviations SE Super-enhancer, TAD Topologically associating domain, CRE Cis-regulatory element, TSS Transcription start site.

retinopathies-associated eQTLs at CREs: $t_{16.603} = -3.6947$, $p = 0.001861$, glaucoma-associated eQTLs at CREs: $t_{4.0371} = -2.2145$, $p = 0.09055$, all eQTLs at SEs: $t_{133.15} = -8.3212$, $p = 9.098\text{e-}14$, retinopathies-associated eQTLs at SEs: $t_{8.0834} = -2.8884$, $p = 0.02003$, glaucoma-associated eQTLs at SEs: $t_{1.0051} = -1.2834$, $p = 0.4206$, Fig. 6F).

Overall, most eQTLs are physically close to their respective eGene, either overlapping the eGene or its promoter, or being in contact through chromatin looping. Our analysis identified multiple potentially functionally-relevant eQTLs overlapping a SE and in physical contact with the promoter of their eGene (e.g., *PRPH2* and *MYO7A* loci; Fig. 6G, top panels) or through chromatin looping (*PRPH2* locus; Fig. 6G, top left panel). We also identified eQTLs overlapping a CRE, either at the promoter of the eGene (e.g., *ABCA1* locus; Fig. 6G, bottom left panel) or by contacting the promoter through chromatin looping (e.g., *PCK2* locus; Fig. 6G, bottom right panel).

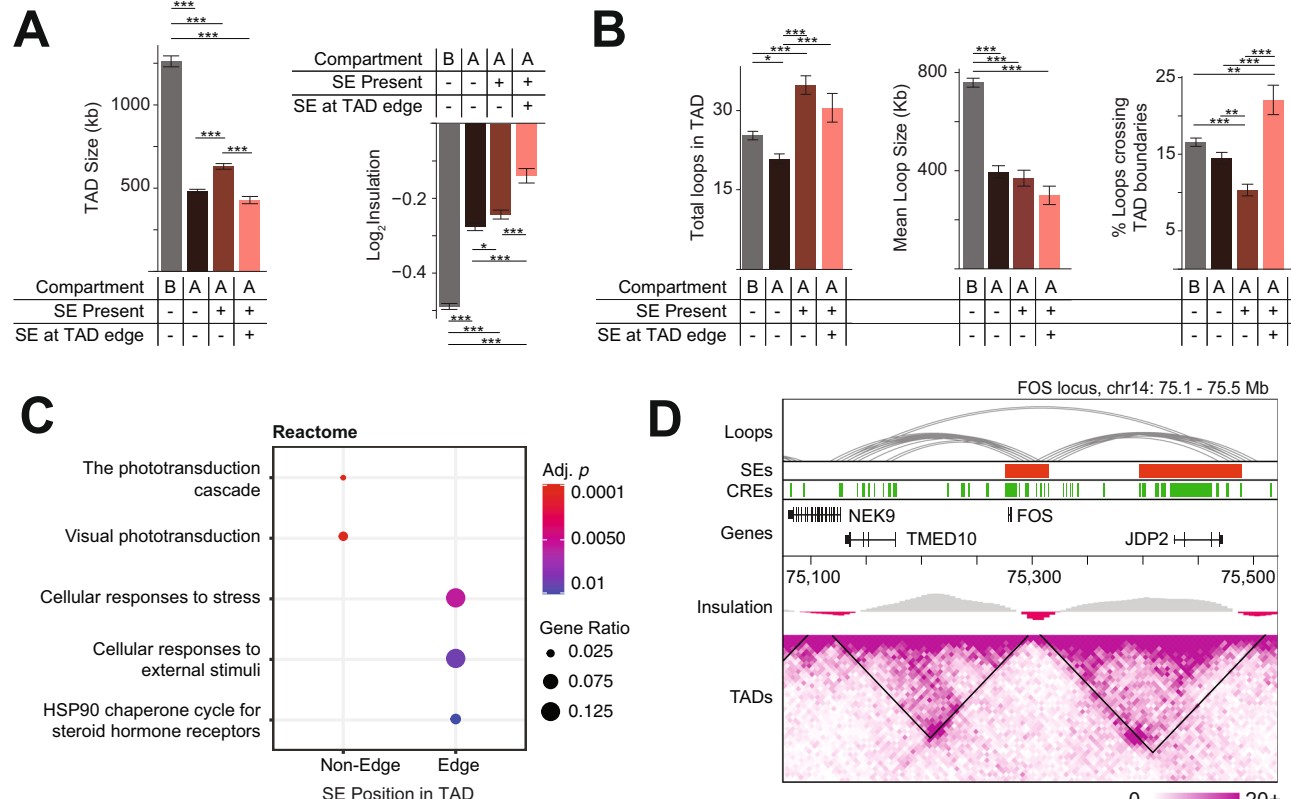

**Fig. 5 | SEs intra-TAD positioning is linked to TADs biological features.** Characterizing TADs fully contained within B compartment or A compartment; A compartment TADs are further divided based upon SE presence/absence and SE position (edge SEs are SEs <5 kb from TAD boundary) (n = 680 TADs in B compartment, n = 670 TADs without SE in A compartment, n = 495 TADs with non-edge SE in A compartment, n = 183 TADs with edge SE in A compartment). These groups vary in [**A**] mean TAD size (two-sided, one-way ANOVA, $F_{2,1557}$ = 270.8, $p < 0.001$) and TAD boundary insulation (two-sided, one-way ANOVA, $F_{2,1548}$ = 91.01, $p < 0.001$; lower $\log_2$ insulation score corresponds to stronger insulation) as well as in [**B**] the number of chromatin loops in contact with the

TAD (two-sided, one-way ANOVA $F_{3,2722}$ = 19.85, $p < 0.001$), mean size of those loops (two-sided, one-way ANOVA $F_{3,2722}$ = 75.47, $p < 0.001$), and percentage of loops which cross TAD boundaries (two-sided, one-way ANOVA $F_{3,2722}$ = 18.66, $p < 0.001$). * indicates $p < 0.05$; ** $p < 0.01$; ***$p < 0.001$ by ANOVA with Tukey HSD post-hoc test. Error bars represent standard error of the mean. **C** Enriched Reactome terms among genes with a TSS residing within or in contact via chromatin looping with a non-edge or edge SE. **D** Chromatin loops, SEs, CREs, $\log_2$ insulation score, TADs, and Hi-C contact maps for the *FOS* locus. *Abbreviations* SE: Super-enhancer, TAD: Topologically associating domain; CRE: Cis-regulatory element; TSS: Transcription start site.

## Genome topology links target genes to AMD- and Glaucoma-associated risk variants

We then leveraged our dataset to investigate chromatin contacts of 52 and 127 lead GWAS variants identified for AMD and glaucoma, respectively[23,24]. We also identified and filtered variants in linkage disequilibrium (LD) with the lead variants (filtered LD variants, see methods)[21] and integrated these with retinal chromatin loops and regulatory elements.

Among AMD variants, 4 lead and 142 filtered LD variants overlap with chromatin loops (Fig. 7A, Supplementary Data 4). Of these, 3 lead variants (C20orf85, CTRB2/CTRB1 and KMT2E/SRPK2 loci) and 136 filtered LD variants distributed among 10 loci [COL8A1, PILRB/PILRA, TGFBR1, ARHGAP21, RDH5/CD63, ACAD10, RAD51B, C3 (NRTN/FUT6), APOE (EXOC3L2/MARK4) and SLC16A8 loci] are in contact with gene bodies or TSS through chromatin loops (Fig. 7A, Supplementary Data 4). In contrast, no overlap is detected between chromatin loops in ACC neurons and AMD lead variants even though 162 of the 196 filtered LD variants overlapping a loop in ACC are in contact with a gene body or TSS. Notably, at the KMT2E/SRPK2 locus, the AMD-associated lead variant rs1142 is in contact with the TSS of KMT2E and SRPK2 and with the gene body of 7 genes, which include KMT2E, SRPK2, EFCAB10, LINC01004, PUS7, AC007384.1 and AC005070.3 through 9 chromatin loops (Fig. 7B, top panel). The lead variant rs72802342, associated with the CTRB2/CTRB1 locus, connects with the gene bodies and TSS of

CFDP1 and AC009054.2 through 6 chromatin loops (Fig. 7B, bottom panel). We also identified 5 AMD lead variants overlapping with a CRE and localizing to gene bodies (ARHGAP21, ARMS2/HTRA1, CNN2, APOE and TNFRSF10A genes) and one AMD lead variant (rs3750846) overlapping with a SE and gene bodies of ARMS2 and BX842242.1 (Supplementary Data 4).

Among glaucoma variants, 12 lead variants and 493 filtered LD variants overlap a chromatin loop (Fig. 7C, Supplementary Data 4). Of these, 8 lead variants (TRAPPC3, LPP, TFAP2B/PKHD1, PDE7B, FBXO32, ADAMTS8, RIC8B and LINC00396/COL4A1 loci) and 363 filtered LD variants are in contact with gene bodies or TSS (Fig. 7C, Supplementary Data 4). In ACC neurons, 10 lead variants and 2,118 filtered LD variants overlap loops; of these, 9 lead variants and 1,890 filtered LD variants are in contact with a gene body or a TSS. For example, the lead variant rs72904286, associated with TFAP2B/PKHD1 locus is in contact with the gene body and TSS of TFAP2B through a chromatin loop (Fig. 7D, top panel). The lead variant rs1037013, associated with RIC8B locus is in contact with the gene body of 4 target genes which include RIC8B, RFX4, POLR3B, and AC079385.1 and with the TSS of RFX4 through chromatin looping (Fig. 7D, bottom panel). We also detected 16 glaucoma lead variants overlapping a CRE with 14 of these localizing to gene bodies or TSS, and 5 glaucoma lead variants overlapping a SE and a gene body or TSS (Supplementary Data 4).

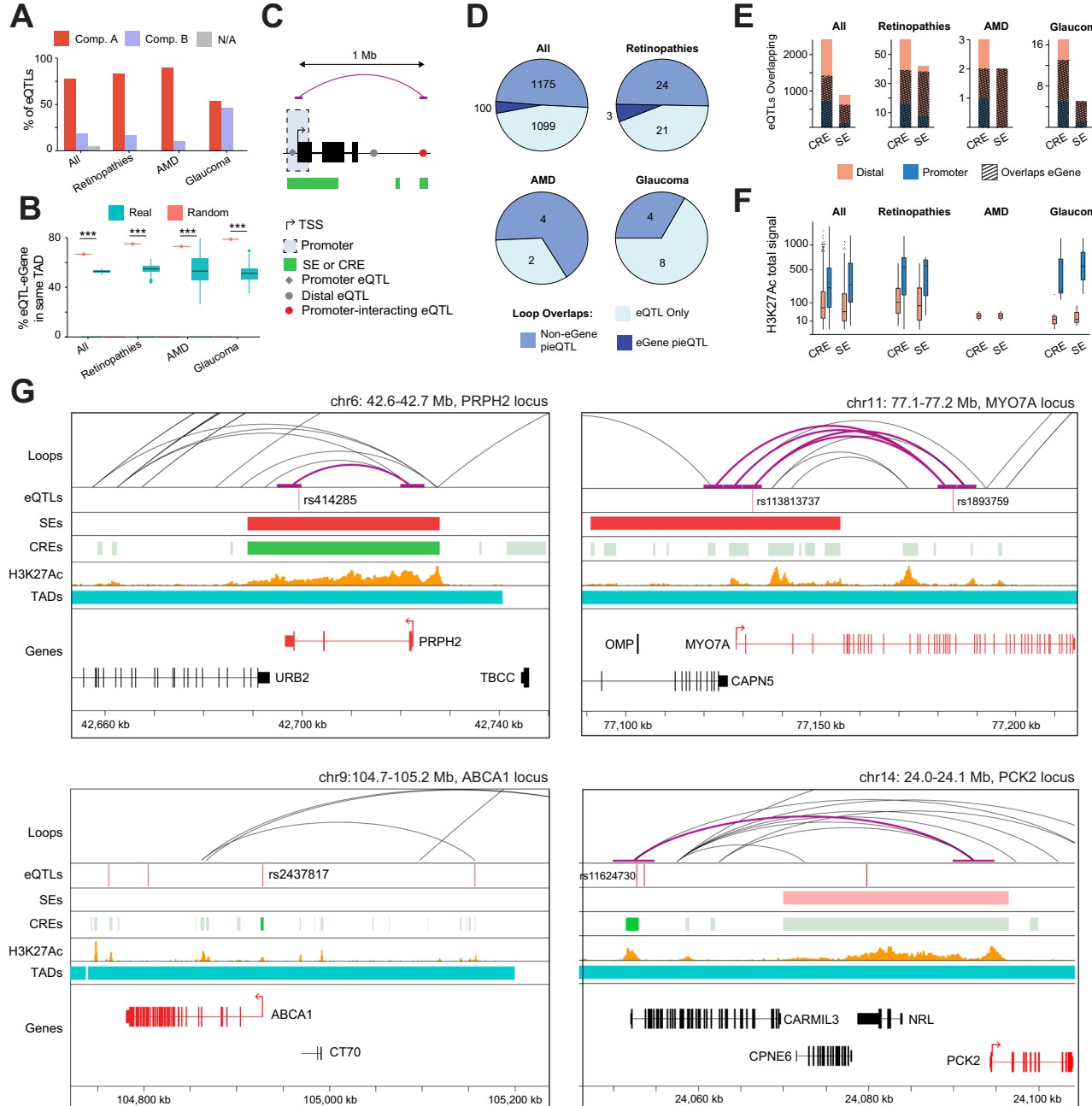

**Fig. 6 | Epigenetic context of retinal eQTLs and eQTLs associated with eye disease.** Proportion of unique eQTLs (variant / eGene pairs) and unique eQTLs involving genes associated with eye disease (**A**) overlapping each chromatin compartment or (**B**) share the same observed ($n = 1$) or random TAD ($n = 100$; see methods). Boxplots represent the median and interquartile range (IQR); whiskers mark 1.5x the IQR; data beyond 1.5x the IQR are plotted as individual points.; *** indicates $p < 0.001$ by two-sided t-test between observed and random TADs. **C** Schematic of eQTL annotation depending on the variant position relative to TSS. **D** Number of unique eQTLs and unique eQTLs involving genes associated with eye disease overlapping with a loop that are not in contact with any TSS (eQTL only), in contact with any TSS excluding the eGene TSS (non-eGene pieQTL), or in contact with the TSS of their eGene (eGene pieQTL). **E** Number of unique eQTLs (variants / eGene pairs) and unique eQTLs involving eye disease genes with the variant overlapping a CRE or SE with plain colors indicating the variant position relative to the eGene TSS and stripes indicating variants overlapping their eGene. **F** Total H3K27Ac coverage around variants (±100 bp) of each unique eQTLs (variants / eGene pairs) and unique eQTLs

involving genes associated eye disease, overlapping a CRE or SE ($n = 1677$ eQTLs overlapping a distal CRE, $n = 744$ eQTLs overlapping a promoter CRE, $n = 12$ glaucoma eQTLs overlapping a distal CRE, $n = 5$ glaucoma eQTLs overlapping a promoter CRE, $n = 44$ retinal disease eQTLs overlapping a distal CRE, $n = 16$ retinal disease eQTLs overlapping a promoter CRE, $n = 2$ AMD eQTLs overlapping a distal CRE; $n = 763$ eQTLs overlapping a distal SE, $n = 121$ eQTLs overlapping a promoter SE, $n = 3$ glaucoma eQTLs overlapping a distal SE, $n = 2$ glaucoma eQTLs overlapping a promoter SE, $n = 34$ retinal disease eQTLs overlapping a distal SE, $n = 8$ retinal disease eQTLs overlapping a promoter SE, $n = 2$ AMD eQTLs overlapping a distal SE). Boxplots represent the median and interquartile range (IQR); whiskers mark 1.5x the IQR; data beyond 1.5x the IQR are plotted as individual points. **G** Examples of promoters and pieQTLs associated to retinopathies (*PRPH2* locus), glaucoma (*MYO7A* locus, top panels), AMD (*ABCA1* locus, bottom left panel) and at *PCK2* locus (bottom right panel). Tracks represent the chromatin loops, eQTLs variants, SEs, CREs, H3K27Ac coverage, TADs and genes.

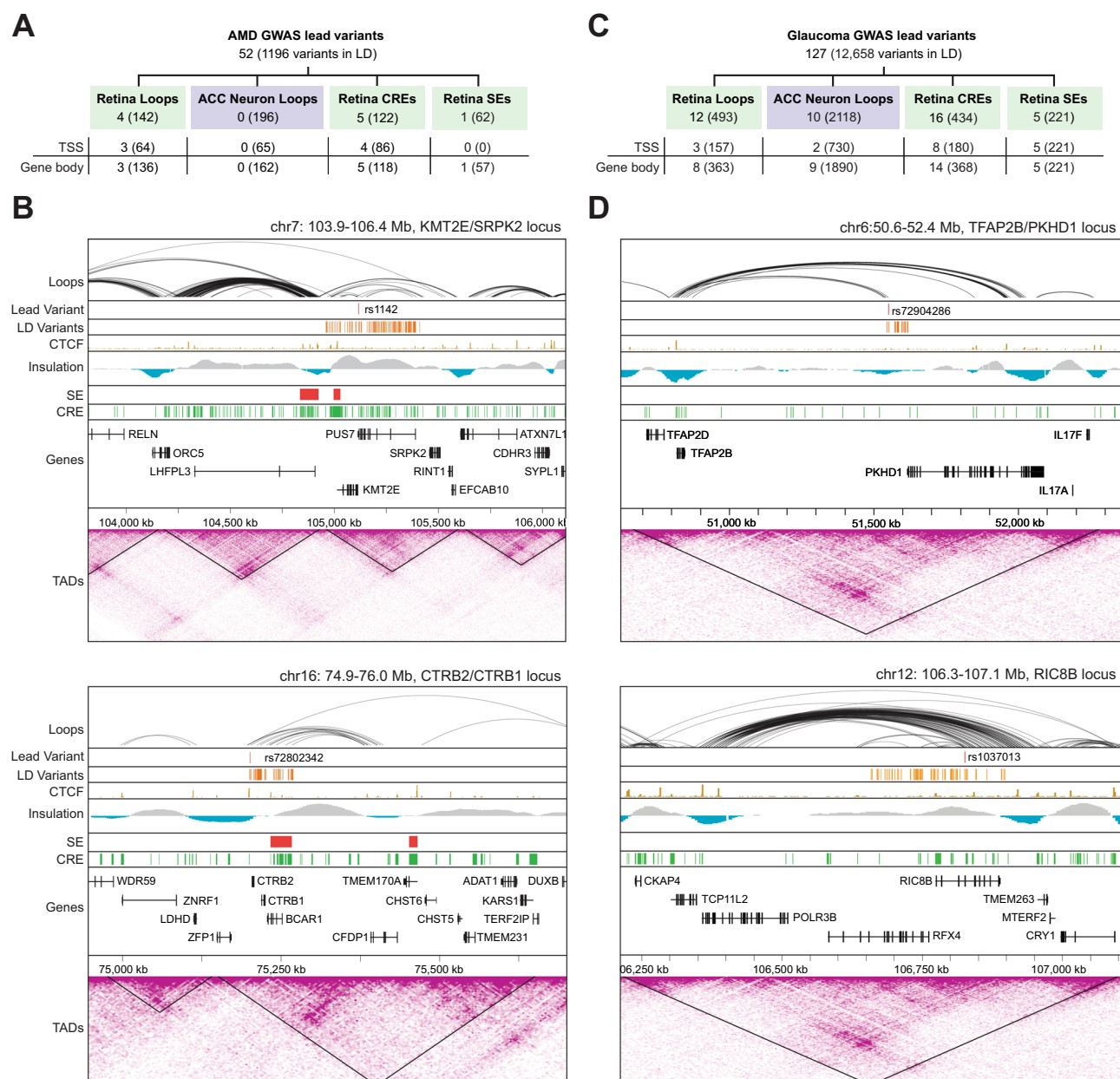

**Fig. 7 | AMD and glaucoma variants are connected to target genes via retinal chromatin loops. A** Count of AMD lead variants and variants in linkage disequilibrium (LD variants; MAF ≥ 1% & R² score ≥ 0.7, shown in parenthesis) contacting genes via chromatin loops in retina or ACC neurons, or residing in retina CREs or SEs. **B** Chromatin loops, lead variants, LD variants, CTCF binding, SEs, CREs, TADs, and Hi-C contact maps for two AMD GWAS loci. **C** Count of glaucoma lead and LD variants (MAF ≥ 1% & R² score ≥0.7, shown in parenthesis) contacting genes via chromatin loops in retina or ACC neurons, or residing in retina CREs or SEs. **D** Chromatin loops, lead variants, LD variants, CTCF binding, SEs, CREs, TADs, and Hi-C contact maps for two glaucoma GWAS loci. Abbreviations AMD Age-related macular degeneration, GWAS Genome-wide association study, LD Linkage disequilibrium, TSS Transcription start site, MAF Minor allele frequency, CTCF CCCTC-binding factor, SE Super-enhancer, TAD Topologically associating domain, CRE Cis-regulatory element.

## Discussion

Spatial architecture of chromatin is highly dynamic and requires active control mechanisms[40]. The adult retina is comprised of non-dividing and highly-specialized sensory neurons and thus offers a relatively stable environment for investigating the role of 3D genome in controlling genetic information. Hi-C allows the exploration of 3D genome architecture by identifying chromatin contacts but can only reveal pairwise interactions. Thus, it is not possible to determine whether several regions interacting together in a Hi-C contact map coexist in each cell of the population, or whether these reflect a heterogeneous cell population each with unique pairs of interacting regions. At higher resolution, fine regulatory structures can resolve such promoter-

enhancer interactions. Our data have a resolution of almost 3 kb, which means that significant chromatin contacts spanning over just few kilobases can be identified. We noted a non-homogeneous average resolution across the genome, with some regions requiring additional sequencing to reach this resolution whereas others exceeding this at our sequencing depth. This heterogeneity was considered when statistically calling significant contacts at each genomic location.

Our deep sequencing of chromatin contacts integrated with chromatin accessibility and histone marks in human retina has provided detailed information on contacts of distal regulatory elements, including SEs, with their cognate promoter regions. Integrating this regulatory 3D map with CTCF and retinal TF binding and CREs[34] has

allowed us to construct a comprehensive gene regulatory network. We show that retina SEs frequently overlap with retina-enriched genes coding for key retinal TFs including NRL, CRX, OTX2, RORB and MEF2D, whereas TFs they encode bind extensively to SEs. This interconnected, self-regulating TF network may represent a core transcriptional regulatory circuit for maintaining cell identity as observed in embryonic stem cells (*sensu*[41]). We have taken advantage of this regulatory architecture to delineate tissue-specific genomic regulation and identify the link between eQTL variants and eGenes via regulatory elements and/or chromatin looping. Finally, by combining our findings with AMD and Glaucoma GWAS, we have uncovered candidate causal genes contributing to these complex traits.

Most SEs we identified are associated with rod genes reflecting rod cell-dominance in human retinal samples. However, we also captured signals from divergent low abundance retinal cell types. For example, relative to random regions, we observe elevated SE overlap with bipolar, cone photoreceptor, and Müller glia genes as well as increased chromatin looping between SEs and horizontal cell genes. Additionally, we noted several retinal pigment epithelial cell markers with TSS overlapping A Compartment and enriched for active regulatory marks (e.g., *BEST-1*, *MERTK*, *RLBP1*, *MITF*, *PMEL*). A previous study of human retina identified accessible chromatin and TF binding peaks assessed by ChIP-seq at regulatory elements of both rod and non-rod genes[34]. In contrast, mostly rod-associated gene states were identified in mouse retina via ChromHMM analysis[42], and no robust chromatin interactions detected between non-rod enhancers/promoters and SE via Hi-C[32]. As predicted, many chromatin interactions correlate with tissue-specific expression and are conserved in mouse.

We demonstrate that the biological function of SE target genes can be associated with SE localization in TADs as well as TAD boundary insulation. Notably, SEs themselves show interaction mostly with loci in close proximity, suggesting strong insulation at the local, sub-TAD level likely due to the enrichment for CTCF binding and to avoid random activation of nearby genes, consistent with previous studies[43]. Unexpectedly, we detect weaker insulation at SE-containing TAD boundaries, in contrast to a previous study showing the association of SEs with increased TAD insulation[44]. We believe this could be due to differing approaches to insulation score calculation. We measure boundary insulation based on contacts crossing a TAD boundary region, as recommended by the 4D Nucleome consortium (i.e., cooltools diamond insulation). In contrast, other study normalized this count using the contacts in adjacent regions[44], leading to a bias of insulation score for adjacent regions rich in local loops (such as the SE-containing regions). We should point out that TAD boundary localization varies at the single cell level[45,46], therefore the insulation score at boundaries can also reflect the heterogeneity in the tissue. Thus, we propose that chromatin architecture could be more dynamic around these transcription hotspots. For example, a SE and its associated genes could be dynamically targeted to, and released from subnuclear compartments, temporally disrupting the TAD boundary in a subset of cells. This would be reflected at the population level by a lower insulation at the affected TAD boundary. In concordance with this hypothesis, SE at the edge of a TAD could affect the boundary to a greater extent compared to SE in the middle of a TAD, leading to a weaker insulation at edge-SE TAD boundaries, as we observe here.

Our integrated analysis has uncovered multi-way SE-chromatin interactions centered around hub-like genomic regions, as observed in other tissues[46–48]. For example, the SE overlapping *FOS* connects large transcriptional units across multiple TADs and may facilitate rapid stress response. These hubs often connect genomic regions across long distances (>250 kb); e.g., contacts between the CREs of *MEF2C*, *VEGFA* and *CLSTN1* with distant genes. Notably, at the *MIR9-2* locus, *MEF2C* interacts with the gene *CETN3* which is associated with retinitis pigmentosa and located over 1 Mb away. This suggests that genetic variations could impact retinal homeostasis and disease by targeting genes over very large distance.

Long-range regulation is facilitated by the physical interaction between regulatory regions and their target genes, providing a probable mechanism for eQTLs to influence expression of eGenes located far from their variants. By integrating retina eQTLs[38] with chromatin looping, SEs, and CREs, we uncovered multiple variants lying in regulatory regions interacting with their target eGene including several pieQTLs, i.e., variants directly contacting the promoter of their eGene. Remarkably, we have identified multiple pieQTLs involving genes associated with retinal neurodegeneration illustrating the value of unbiased genetic association studies to identify genes linked to retinal diseases. Examining the chromatin state at these eQTLs can help prioritize specific variants for further investigation. For example, the variant associated with *PCK2* is lying in a large CRE enriched for active histone marks. Despite being almost 100 kb away from *PCK2*, it is in direct contact with its promoter through a chromatin loop. Thus, different alleles at this locus could impact *PCK2* regulation affecting CRE efficiency, TF binding, and/or chromatin loop formation or stability.

Retinal CREs and eQTLs having high resolution chromatin contacts should permit systematic analysis of relevant non-coding regulating regions for missing heritability and help in addressing issues of variable penetrance in inherited retinal diseases[26]. Rare coding variants in over 200 genes associated with photoreceptor and/or retinal pigment epithelium (RPE) function can lead to vision impairment (RetNet; https://sph.uth.edu/retnet/); yet, only in few instances, non-coding or structural alterations have been associated with human retinal disease[34,49]. Variants in regulatory elements have been implicated in altering the effect of coding mutations on phenotypes[27,50]. We propose that the analysis of functionally-relevant non-coding regions in rhodopsin, *ABCA4* and other known retinopathy genes, as identified in this study, would greatly augment our understanding of Mendelian retinal diseases.

Adult-onset multifactorial diseases affecting retinal function are the major cause of irreversible vision impairment in humans. GWAS of Glaucoma and AMD have identified a large number of non-coding variants, and additional studies including eQTL analysis have provided further insights; yet, the causal genes and variants continue to be elusive for many associated loci. Our integrated analysis of adult human retinal genome topology with GWAS lead and LD variants has uncovered several previously unidentified genes potentially contributing to glaucoma and AMD. For example, we show a remarkable long-range interaction leading to identification of the target gene *TFAP2B* for glaucoma-associated variant rs72904286 at the *TFAP2B/PKHD1* locus. Disruption of *TFAP2B* expression in mouse results in a strong pathologic phenotype consistent with glaucoma[51,52]. Interestingly, this phenotype is believed to be due to defects in the eye tissues originating from the neural crest[52], while we identified *TFAP2B* through data from the retina, originating from the neuroepithelium. It is possible that this apparent discrepancy is due to a lack of tissue specificity of the regulatory region containing the variant rs72904286, which could form non-tissue specific contacts with *TFAP2B*. Indeed, we demonstrate a similar number of chromatin interactions among glaucoma associated eQTLs and risk variants in both retina and in neurons, demonstrating weak tissue specificity for this eye disease. In contrast, AMD-associated eQTLs and risk variants reveal a strong enrichment for chromatin looping in the retina compared to neurons, demonstrating the value of examining genomic architecture of disease variants in the affected tissue.

Our study has also led to identification of candidate causal genes for AMD. For example, we establish *CFDP1* as a candidate gene involved in AMD at the *CTRB2/CTRB1* locus. While the lead variant found at this locus (rs72802342) was previously associated to the target gene *BCAR1*[53], we show association of filtered variants in LD with rs72802342 to the gene coding for *CFDP1*. Indeed, the LD region at this

lead variant is overlapping a retinal SE (chr16:75233500-75267000), which interacts with *CFDP1* through chromatin looping. We suggest that this variant could alter the expression of *CFDP1* by disrupting the SE structure and the interactions it facilitates. This hypothesis is strongly supported by the presence of a variant (rs11641532) in this LD region previously linked to the expression of *CFDP1* in TWAS analysis[23,38]. Conservation of this genomic region from zebrafish to mammals[54] further underscores the importance of regulatory elements contained within. Notably, zebrafish mutants of *Cfdp1* exhibit a loss of Neurod1 positive cells in the retina[55]. Altogether, this places regulation of *CFDP1* expression in the retina as a strong candidate for AMD risk. This example demonstrates how the integration of regulatory elements to 3D chromatin interactions can help clarify the biological impact of variants associated with retinal, and more broadly, human diseases. Notably, one limitation of the Hi-C approach is that the physical interaction between candidate enhancers and promoters does not directly demonstrate a functional relationship with respect to gene regulation. Additional studies are necessary to validate the significance of chromatin interaction on retina-specific gene expression.

In summary, we have generated a significant resource for the human retina, by integrating high resolution Hi-C data with epigenetic profiles and CREs, thereby facilitating investigations of genomic regulation, identification of missing heritability in retinopathies, and candidate causal genes and variants for common blinding diseases including AMD and glaucoma. Our studies thus provide a framework for connecting regulatory variants with retinal disease phenotypes and may assist in design of targeted translational paradigms by modulating genomic regulation.

## Methods

### Retina tissues
Five postmortem human donor eyes (from 2 females and 3 males of 65–77 years age at the time of death) were procured from The National Disease Research Interchange (Philadelphia, PA) (protocol DSWAS 001). Autopsy material from unidentified deceased persons is excluded from review by Institutional Review Board and does not require an Office of Human Subjects Research Protections (OHSRP) determination per 45 CFR 46 and NIH policy (OHSRP ID#: 18-NEI-00619). Eyes were enucleated within 14 h of death and stored in Dulbecco's Modified Eagle Medium (DMEM) (Thermofisher, Waltham, MA) supplemented with antibiotics at 4 °C until dissection. Retinas were dissected and divided into four regions (dorsal, ventral, nasal and temporal) for further processing. Dorsal regions were used in further experiments.

### Hi-C experiment
Freshly dissected human retina tissue (from 3 male and 1 female of 75–77 years age at the time of death) was crosslinked with 1% formaldehyde in PBS for 10 min, quenched with 125 mM glycine for 5 min and frozen until use for Hi-C experiments. Before processing for Hi-C, samples were crosslinked again in 2% formaldehyde for 10 min, then quenched with 125 mM glycine 5 min at room temperature. Hi-C was performed on isolated nuclei using Arima-Hi-C kit (Arima Genomics, CA, USA) and Hyper Prep DNA-seq library prep kit (KK8502; Kapa Biosciences, MA, USA), following the manufacturers' instructions. For HCT116 cell line (ATCC, VA, USA), $7 \times 10^6$ cells were crosslinked with 2% formaldehyde for 10 min, quenched for 5 min with stop solution I, lysed, and processed for Hi-C using ARIMA-Hi-C kit. All libraries were sequenced on HiSeq 2500 platform (Illumina, CA, USA) at a read length of 101 to 126 base pairs with a depth of approximately 300 million read pairs per sample for retina and 50 million read pairs for HCT116.

### Cleavage under targets and release using nuclease (CUT&RUN)
Peripheral retina sample (from 1 female of 75 years age at the time of death) was dissociated using papain as previously described[56], cryo-preserved in HBSS solution containing 10% DMSO and slowly frozen

using a Mr. Frosty container (Invitrogen, CA, USA). CUT&RUN was performed as previously described[57] using 200,000–300,000 cells per experiment. Briefly, cells were bound to activated concavalin A beads for 7 min at room temperature. Antibodies against H3K9me3 (Rabbit, cat.no. ab8898, Abcam, Cambridge, UK), H3K4me3 (Rabbit, cat.no. ab8580, Abcam, Cambridge, UK) and control IgG (Rabbit, cat.no. 011-000-002, Jackson ImmunoResearch Laboratories, PA, USA) were used at a concentration of 1:100 in 100 µl overnight at 4 °C. pA-MNase conjugated to protein A (generous gift of Dr. Steven Henikoff, Howard Hughes Medical Institute, WA, USA) was used at a concentration of 700 ng/ml for 1 hr at 4 °C and activated with calcium chloride for 30 mins at 0 °C. Released DNA fragments were purified using QIAquick PCR Purification Kit (QIAGEN, Hilden, Germany). Libraries were prepared with SMARTer® ThruPLEX® DNA-Seq Kit (Takara Bio USA, Inc, CA, USA) and amplified with 15 PCR cycles (60 °C extension). Pair-end sequencing was performed with read length of 50 base pairs using the HiSeq 2500 platform (Illumina, CA, USA).

### Assay for Transposase-Accessible Chromatin using sequencing (ATAC-seq)
ATAC-seq was performed using fresh dissociated cells from five retinas (2 females and 3 males of 65–77 years age at the time of death). Tagmentation and library preparation were carried out as described[58] using Nextera DNA Library Prep kit (FC-121-1030, Illumina, San Diego, CA). Dissociated cells were quantified using a hemocytometer, and the nuclear fraction of 50,000 cells was incubated with tagmentation reaction mix (5 µl Tn5, 25 µl 2x tagmentation buffer, 20 µl nuclease-free water) in a thermomix with 600 rpm agitation. The DNA was purified by "MinElute PCR Purification" kit (Qiagen, 28004) followed by a two-sided size selection using a 0.5 and 1.5 ratio of SPRIselect reagent (Beckman Coulter B233181). All of the DNA was used to prepare the libraries, which were sequenced pair-end using the HiSeq 2500 platform (Illumina, CA, USA) at a read length of 50 base pairs.

### Hi-C data processing
Hi-C analysis was performed with HiCUP v0.8.0[59] using Arima-specific in silico digested hg38 genome for the retina and HCT116 samples, and MboI in silico digested hg38 genome for the published neuron and GM12878 samples[2]. Filtered bam files produced by HiCUP v0.8.0[59] were converted to .hic files[60] and to HOMER[61,62] tag directories. Retina Hi-C resolution was estimated at 3.06 kb using HiCRes[63]. Reproducibility between samples as well as correlation with Hi-C from other tissues were assessed using HiCRep v1.0.0[64]: by computing the average SCC for chromosomes 1 to 22. Compartments were called with Homer v4.1[61,62] using 100 kb sliding windows with a step of 50 kb (using options superRes and res, respectively). Compartment correlation between different tissues was computed using R. Loops were called using FitHiC v2.0.7[65] on the merged Hi-C datasets, using 5 kb resolution and an FDR threshold of 0.01. TADs calling was performed on the merged samples with domaincaller[66], using raw contact maps of 10 kb bins. Insulation scores were computed using the diamond insulation tool from cooltools v0.4.0, using 100 kb windows with 5 kb bins.

### ATAC-seq data processing
ATAC-seq from five human retinas was analyzed by trimming the reads for Nextera transposase sequence and reducing their size to maximum 25 bp using cutadapt v3.0[67] and FASTX toolkit (FASTX-Toolkit v0.0.14, *RRID:SCR_005534*). Trimmed reads were mapped on hg38 using bowtie2 v2.3.5.1[68] and filtered for quality, duplicates and blacklisted regions (ENCODE dataset ENCFF419RSJ) using samtools v1.9[69]. Two sets of aligned read pairs were produced: TN5-shifted and non-shifted, to use with general tools (TN5-shifted) or specialized tools already including a TN5 shifting step (non-shifted). Quality controls were performed on

the aligned read pairs by computing the normalized enrichment at TSS using HOMER[61,62], and by computing the fragment size. Retina 1 was chosen as the best dataset from this QC and used as input for ChromHMM[33] annotation v1.19 (see Chromatin annotation section).

## Cut&Run processing

Cut&Run datasets from one retina were processed by trimming the adaptors using cutadapt v3.0[67] and mapping locally with bowtie2 v2.3.5.1[68], allowing dovetail on a chimeric genome (Human hg38, *S. cerevisiae* S288C; *E. coli* ASM584v2). No internal normalization was performed in these datasets since the proportion of read mapping on the yeast or on the bacterial genome was too low. Reads mapping on the human genome were then extracted, filtered for quality, duplicates, and blacklisted regions (ENCODE dataset ENCFF419RSJ) using samtools v1.9[69]. Quality of the experiment was visualized on IGV v2.11.9[70] after conversion to bigwig using deepTools[71] (Supplementary Fig. 6).

## ChIP-seq data processing

Public ChIP-seq datasets from human retina (GSE137311[34]) were reanalyzed using the same parameters as for Cut&Run, i.e., locally mapping the reads and allowing dovetail. Mapped reads were filtered for quality, duplicates, and blacklisted regions (ENCODE dataset ENCFF419RSJ) using samtools v1.9[69]. To select only one dataset per chromatin mark, autocorrelation was assessed using HOMER[61,62] and datasets with the highest same and different strand enrichments and the ratio same / different strand fold enrichment closer to one were selected for further analysis. Global quality was visually assessed using IGV v2.11.9[70] on mapped reads converted to bigwig using deepTools v3.3.0[71] (Supplementary Fig. 6). Chromatin binding peaks of CREB, CRX, CTCF, MEF2D, NRL, Otx1/2 and RORB were identified using MACS2 v2.2.7.1[72].

## Tissue- and cell type-enriched genes

Tissue-enriched genes and cell type-enriched gene lists used in Figs. 2C, 3E and Supplementary Fig. 4 were downloaded from the Human Protein Atlas[73].

## Chromatin annotation

The chromatin annotation used for this project is an integration of public (H3K4me2, H3K27Ac) and lab-produced (H3K4me3, H3k9me3) chromatin histone marks with our best chromatin accessibility dataset (see ATAC-seq processing section). This integration was performed by transforming the read coverage to 500 bp signal and computing chromatin states using the ChromHMM[33] tool v1.19. Models defining between 5 and 20 chromatin states were tested. For each model, we calculated the mean correlation between each chromatin state and the most similar chromatin state in each model with additional states. After 10 chromatin states, this correlation plateaued indicating that additional chromatin states provide minimal information. For each state of the 10-states model, the average coverage for the chromatin marks and accessibility were computed using HOMER[61,62] and the chromatin signatures for each segment were used to manually infer a biological annotation of each state. The average distributions of each state around the expressed and non-expressed genes were computed as a quality control using HOMER[61,62], with our previously published list of expressed genes in human retina[38] and all the other genes from HOMER TSS list as non-expressed genes.

## CREs elements and SEs identification

Chromatin states enriched for active histone marks identified by ChromHMM v1.19 were merged to generate a set of regions called here CREs. This list of CREs was integrated with the H3K27Ac ChIP-seq coverage and the corresponding input coverage to identify SE using the ROSE algorithm[20,74].

## SE characterization

Basic SE characteristics were computed using R. Correlation between SE count and chromosome size or number of retina-expressed genes is measured as Pearson's $r^2$ computed with ggpubr v0.4.0. To identify statistically significant enrichment in SE characteristics, 100 random region datasets have been generated using bedtools random v2.29.2[75] with a length corresponding to the median length of SE and the number of regions per dataset equal to the number of SE. SEs and random regions have been overlapped with the Hi-C loops using bedtools PairToBed v2.29.2[75]. Statistics of the overlaps were computed using R. SEs and random regions were overlapped with all expressed genes as well as tissue- and cell type-enriched genes from the Human Protein Atlas[73] using bedtools intersect v2.29.2[75]. Genomic regions in contact with a SE or random region through chromatin were also overlapped with tissue- and cell type-enriched genes, CREs, and heterochromatin regions (from chromHMM[33] annotated states) using bedtools intersect v2.29.2[75].

## Accessible motifs analysis

To identify accessible motifs, accessible chromatin loci were identified using MACS2 v2.2.7.1[72] on each dataset and the average coverage at ATAC peaks has been computed as a quality control using HOMER[61,62]. Footprints have been extracted from each list of accessible loci using rgt-HINT v0.13.1[76]. Accessible footprint loci less than 20 bp apart have been merged for all ATAC-seq experiments. Loci found in at least 3 retinas have been kept for motif finding. To control the quality of accessible footprint discovery, we identified binding motifs present within these footprints at an FDR < 0.05 using FIMO[77] with the motif database HOCOMOCOv11[78]. Then, the read coverage from CRX, NRL, CTCF and Input ChIP-seq around each CRX, NRL or CTCF accessible motifs were computed using HOMER and plotted using R.

Enriched TF binding motifs from Fig. 3H were computed by running AME from MEME suite v5.4.1 on accessible footprints within SE and containing CRX and NRL binding sites. SE-overlapping footprints without NRL or CRX binding sites were used as background. Finally, motifs from expressed TFs were filtered to those expressed in a previous retina transcriptome study (GSE115828).

## Characterization of retinal eQTLs

The 14,859 eQTLs identified previously[38] were classified based on the location of the variants relative to the canonical TSS of the associated eGene from Ensembl 102. Expression quantitative trait loci variants located within ±2.5 kb of the eGene TSS were identified as promoter eQTLs while those located >2.5 kb from the eGene were identified as distal eQTLs. One hundred sets of random TADs of the same count and mean length as the real TADs were generated using regioneR v1.26.1[79]. For each set of TADs, real and random, we determined what proportion of eQTL variants resided within the same TAD as their associated eGene TSS. Next, eQTL variants which overlap a chromatin loop foot were classified based on the location contacted by opposite loop foot; variants in contact with their eGene promoter were identified as eGene pieQTLs, variants in contact with a promoter other than the eGene as non-eGene pieQTLs, and variants not in contact with any promoter as eQTL only. Finally, eQTLs were checked for overlap with CREs and SEs then subsequently checked if those regions overlapped the associated eGene. All overlaps were performed using GenomicRanges v1.42[80] in R.

## Identification of linkage disequilibrium (LD) variants for AMD and Glaucoma GWAS loci

Fifty-two AMD associated variants distributed across 34 loci and 127 glaucoma associated variants and loci were considered for the analysis[23,24]. We calculated linkage disequilibrium (LD) for 52 AMD lead genetic variants and for 127 glaucoma lead variants within 1 MB using LDlink v5.1, among individuals with Europe ancestry from 1000 Genomes Project data[21,81]. Using hg38 coordinates, we identified 65,625

AMD variants and 188,465 glaucoma variants in LD; of these, 1,196 filtered AMD variants and 12,658 filtered glaucoma variants in LD with MAF ≥ 1% and $r^2 \geq 0.7$ were selected for further analysis.

## Identification of AMD and Glaucoma GWAS loci target genes

The closest target genes overlapping with loops, CRE, SE, 52 AMD lead variants, 1,196 filtered AMD LD variants, 127 glaucoma lead variants and 12,658 filtered glaucoma LD variants were obtained using the closestBed command from bedtools v2.29.2[75]. Gene and TSS hg38 coordinates from Ensembl version 102 were used to overlap with the coordinates of the 52 AMD lead variants, 1,196 filtered AMD LD variants, 127 glaucoma lead variants and 12,658 filtered glaucoma LD variants. For a loop target gene, one foot of the loop overlaps the AMD/glaucoma GWAS lead variants or filtered AMD/glaucoma LD variants, and the second foot of the loop overlaps the gene body or TSS of a gene. CRE and SE target genes were defined by both the AMD/Glaucoma GWAS lead variant (or filtered AMD/glaucoma LD variant) and the gene body (or TSS of a gene) overlapping the same CRE.

## Figure plots

Contact maps were plotted using the Washington University Epigenome browser[82] or using HOMER[61,62] and R v3.6 and v4.0.3. Chromatin profiles and chromatin loops were plotted using IGV v2.11.9[70]. Graphs were plotted using R v3.6 and v4.0.3, ggplot2 v3.3.5 and dplyr v1.07 and ComplexHeatmap v2.7.10.9002.

## Reporting summary

Further information on research design is available in the Nature Research Reporting Summary linked to this article.

## Data availability

The data that support this study are available from the corresponding author upon reasonable request. Datasets produced in this study are accessible in GEO under the accession numbers: GSE202471 (Hi-C), GSE202472 (ATAC-seq), GSE202473 (Cut&Run) and GSE202474 (full series). These data can be explored using our user-friendly application on computer, tablet, or smartphone on http://grn.nei.nih.gov. hg38 genome was used for alignment. Gene expression data are from our previous study, under the accession GSE115828[38]. The public ChIP-seq raw data used in this study are accessible under the following SRA numbers[34]: SRR10172858 (H3K27Ac), SRR10172898 (H3K4me2), SRR10172903 (CRX), SRR10172897 (NRL), SRR10172909 (CTCF), SRR10172910 (MEF2D), SRR10172908 (RORB), SRR10172914 (CREB), SRR10172882 (OTX1 / OTX2), SRR10172850 (Input). Public Hi-C data are accessible under the following accession numbers: GSE135465 (Mouse retina[32]), Synapse syn12978758 and syn12978762 (Purified neurons[31]) and SRR1658572 (GM12878[2]).

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

## Acknowledgements
We are grateful to Freekje van Asten and Benjamin Fadl for help in dissection of retinal tissue, Linn Gieser for assistance with next generation sequencing, and Matthew Brooks, Laura Campello Blasco, Kamil Kruczek, Anupam Mondal and Zepeng Qu for helpful discussions. This work was supported by Intramural Research Program of the National Eye Institute (ZIAEY000450 and ZIAEY000546) and utilized the high-performance computational capabilities of the Biowulf Linux cluster at NIH (http://biowulf.nih.gov).

## Author contributions
Overall Conceptualization, C.M., N.S., X.C.D., and A.S.; Experimental work, N.S., C.J., and X.C.D.; Data analysis, C.M., Z.B., and J.A.; writing original draft, C.M., N.S., Z.B., J.A., X.C.D., and A.S.; Editing, all authors; Funding Acquisition, Supervision, and Project Administration, A.S.

## Competing interests
The authors declare no competing interests.
