## [Peer Review File · Nature Communications]

REVIEWER COMMENTS

Reviewer #1 (Remarks to the Author):

This study generated a high-resolution genome topology of human retina by Hi-C. The authors characterized super-enhancer-associated chromatin 3D organization and their functions. In addition, the authors also revealed the relationship between chromatin loops and eQTL-eGene. Although the Hi-C database resource is important for the future study of retina, this study lacks deep exploration and functional experiments. The major concern is that all the work in this manuscript is descriptive and lacks novelty. Most of the results in this study are unimportant, predictable and well-known, especially Figure 1,2,3,4. Only few interesting findings or novel regulatory patterns were proposed in this manuscript.

Some additional concerns are as follows:

1. In line 233-234, the authors indicated that “TADs with edge SEs are enriched for stress response genes suggesting the need for more dynamic and transitional interactions.” The observation that “TADs with edge SEs are enriched for stress response genes” can’t make the conclusion that “stress response genes need for more dynamic and transitional interactions”. More evidences are needed to verify this conclusion.
2. The authors analysed the overlapping between AMD- and Glaucoma-associated risk variants and chromatin loops. Functional experiments should be done to explore whether the GWAS variant-associated interactions is associated expression of target genes. Can the disruption of these chromatin interactions lead to AMD- and Glaucoma-associated gene expression changes and phenotype?
3. The authors should explain what is “regulatory hubs of SEs”?
4. In Supplementary Fig 3E, unit of y-axis did not been indicated.
5. In line 168, figure 3B is an error indication. It should be figure 3C. In line 173, it should be figure 3B not 3C.
6. What is the meaning of the label “% Gene w/ TSS in SE” in Figure 3E? please clarify it.

Reviewer #2 (Remarks to the Author):

In this work, Marchal, Singh, Batz et al. introduce a new, high resolution Hi-C dataset and other genomics data generated from adult human retinal tissue. They integrate these new data with existing datasets to define and characterize topologically associated domains (TADs) and chromatin loops. They benchmark their new data by comparing results between biological replicates and across different cell types. Then, they compare TAD boundaries with H3K27ac-defined super enhancers, gene expression, eQTL analyses, and GWAS loci. From their analyses, the authors conclude that TAD insulation is stronger if a super enhancer is located centrally rather than near the edge of that TAD and that functional links between variants and their corresponding genes are revealed by merging Hi-C with eQTL or GWAS analyses.

In this reviewer's opinion, the major strength and noteworthy result of this work is the generation and quality assessment of the novel genome-wide HiC dataset. The authors' integrative approach convincingly demonstrates the quality of these data and their potential for addressing the role of chromatin conformation in regulation of retinal gene expression. By making this resource available, the authors fill a major gap in the field, enhance the ability to pair regulatory elements with target genes, and advance efforts to interpret non-coding variants in the context of retinal disease. I would suggest the following recommendations to increase the impact of the work:

Major suggestions:

1) I would encourage the authors to make this important resource as accessible as possible for other researchers, for example by providing ".hic" files for visualization of contacts in applications like Juicebox, the UCSC genome browser, or equivalent. Similarly, facilitating easy visualization of chromatin loops would allow for informal browsing of the data, which will likely allow more users to appreciate and build upon this important work. For the same reason, please provide a table of identified super enhancers in a ".bed" or equivalent format.

2) The authors should consider revising some statements regarding "functionality" since strictly speaking the analyses presented here are based on correlations and associations. The authors could strengthen the connection of this work to biological function by performing further experiments, but I believe that would be beyond the scope of this manuscript. I don't think it would warrant delaying the publication of these data.

3) I expect this work will be broadly read and cited, so I think a frank assessment of the strengths and weakness of HiC data in the Discussion is warranted. For example, there is a commonly held idea that chromatin conformation data can be used to unambiguously pair enhancers with their functional targets. Given the depth and resolution of their data, I would like the authors to comment on to what degree this is possible and what limitations remain. A very brief mention of the technical limitations and potential artifacts of HiC should also be included.

4) Several figures in the paper make conclusions from quantitative data without reporting specific statistical analysis. The authors should perform and report the appropriate statistics where relevant. Such panels include 1E, 2C, 2D, 2E, 2F, 4A, 4C, 4D, 6A, 6B, 6E, and 6F.

Minor comments:

1. The authors use the NRL locus as an example in Figure panels 1D, 3F, and 6G. In each case, it appears that a different isoform of NRL is shown. Please clarify which isoform is most representative for the adult human retina or why different isoforms are shown in each instance.

2. Please clarify in the online methods the number of retinas from female and male donors that were used for the HiC vs the CUT&RUN analyses.

3. There are a few grammatical and typographical errors throughout the manuscript that should be addressed, including:

a. lines 68-71 is missing a preposition between “little” and “half”

b. Panels 3B and 3C are switched in figure 3 relative to the text in lines 168-173

c. Line 522 “Read mapping on the human genome” should be “Reads mapping on the human genome”

4. Lines 141-144 describing 2F are somewhat confusing and may benefit from being split into two sentences.

5. The authors should clarify why they are comparing raw contact counts in Figures 2D and 2F. Would this be expected to be biased according to sequencing depth differences between the datasets?

Detailed Response to the Reviewers

This study generated a high-resolution genome topology of human retina by Hi-C. The authors characterized super-enhancer-associated chromatin 3D organization and their functions. In addition, the authors also revealed the relationship between chromatin loops and eQTL-eGene. Although the Hi-C database resource is important for the future study of retina, this study lacks deep exploration and functional experiments. The major concern is that all the work in this manuscript is descriptive and lacks novelty. Most of the results in this study are unimportant, predictable and well-known, especially Figure 1,2,3,4. Only few interesting findings or novel regulatory patterns were proposed in this manuscript.

We thank the reviewers for providing valuable comments. We agree that the study lacks functional experiments. However, the goal of the current manuscript is to provide a chromatin contact map of human retina as a useful resource and characterize basic principles of gene regulation in the human retina by integrating new and published datasets including eQTLs and GWAS. Functional experiments based on such a global genome topology map will require specific biological focus and are beyond the scope of this study. We wanted to make this resource available to the community in a timely manner. As reviewer 2 highlights, we expect this manuscript to be widely used and highly citable. Anecdotally, the senior author has presented this work recently in a keynote address and the response was universally positive, highlighting the appetite for this dataset among vision researchers.

We respectfully disagree with this reviewer's assessment of our manuscript's novelty. Our study generated the first high resolution (3-5 kb) contact map of the human retina (and is among only a couple of others for human tissues). We identify retina specific super enhancers and the multiway chromatin contacts they form within rods and other cell types. Furthermore, we integrated the genome topology data with retinal eQTLs as well as GWAS for AMD and glaucoma, providing novel insights into causality.

We also disagree about the value of figures 1-4 as they provide information on data quality, conservation of human and mouse genomic architecture, novel candidate retinal transcriptional regulators, and key super-enhancers that regulate multiple genomic loci in the retina. While some of the results in these figures seem "predictable", the field of chromatin topology is still very new and there are emerging contradictory findings on many aspects of chromatin architecture that need to be resolved; including whether chromatin topology is highly conserved (see e.g., PMID: 33203573). Our study of a human tissue thus offers further insights into this important field. We must add that predictions require evidence before acceptance. This work provides the first high quality integrated dataset for testing predictions about human retina genomic architecture and its relationship with gene regulation.

1. In line 233-234, the authors indicated that "TADs with edge SEs are enriched for stress response genes suggesting the need for more dynamic and transitional interactions." The observation that "TADs with edge SEs are enriched for stress response genes" can't make the conclusion that "stress response genes need for more dynamic and transitional interactions". More evidence are needed to verify this conclusion.

We agree with the reviewer's comment. Perhaps, we were not clear in our statement. The sentence "TADs with edge SEs are enriched for stress response genes suggesting the need for more dynamic and transitional interactions" is an observation and no conclusions were made, although we do hypothesize about the possible cause of this observation in the discussion

section. To limit speculations, we have changed “suggesting” to “indicating” in the revised manuscript (line 237).

2. The authors analysed the overlapping between AMD- and Glaucoma-associated risk variants and chromatin loops. Functional experiments should be done to explore whether the GWAS variant-associated interactions is associated expression of target genes. Can the disruption of these chromatin interactions lead to AMD- and Glaucoma-associated gene expression changes and phenotype?

We must reiterate that AMD and Glaucoma are multifactorial aging-associated diseases involving variants at multiple genetic loci and influenced by environmental factors. GWAS provides associated genomic loci and not causal genes. For AMD, at least 34 genetic loci have been confirmed and for glaucoma, this number is over 100. GWAS-identified variants are largely present in non-coding regions and likely impact gene expression patterns as indicated by our previous eQTL study (Ratnapriya et al. Nat Genet 2019). Our current study links retinal eQTLs and GWAS-identified lead and LD variants (primarily in non-coding genome) to candidate causal genes for AMD and glaucoma. We would like to point out that we have identified interactions observed in independent genomic association analysis (e.g., eQTLs), in a way indicating functional interactions.

We note that contact maps presented in our manuscript show interactions in the retina-expressed genes. Genome topology at retina-specific genes is not observed in other tissues/cells but is conserved in the mouse retina.

Expression changes are predicted by disruption of chromatin interactions but are unlikely to provide a disease phenotype as AMD and glaucoma are multifactorial and age-dependent. We should note that currently available model systems are not ideal for testing complex trait phenotypes.

We agree that these functional experiments are important. However, such experiments will take years to complete since one will need to develop appropriate model systems and, in our opinion and as mentioned by reviewer 2, are beyond the scope of this study. At this stage, we have initiated projects focusing on specific questions to further characterize our findings, and those studies would be subject of future manuscripts. Making this data available quickly to the community will expedite validations by different research groups help move the field forward.

3. The authors should explain what is “regulatory hubs of SEs”?

This phrase was intended to describe clusters of super enhancers in contact with one another via chromatin looping. We have revised the text in the manuscript (lines 215-217) to communicate that more clearly.

4. In Supplementary Fig 3E, unit of y-axis did not been indicated.

We have revised Supplementary Figure 3E so the y-axis now reads “Number of SEs” instead of “SEs”.

5. In line 168, figure 3B is an error indication. It should be figure 3C. In line 173, it should be figure 3B not 3C.

We apologize for this error. We have now correctly cited figure 3B and 3C in the revised manuscript.

6. What is the meaning of the label “% Gene w/ TSS in SE” in Figure 3E? please clarify it.

Panel 3E shows the percent of genes in various enrichment groups (e.g., retina-enriched genes, brain enriched genes, etc.) that have a transcription start site overlapping with a SE or random SE-sized region. We have updated the label to read “% Genes Overlapping” and revised the legend to clarify the meaning (lines 879-880).

Reviewer #2 (Remarks to the Author):

In this work, Marchal, Singh, Batz et al. introduce a new, high resolution Hi-C dataset and other genomics data generated from adult human retinal tissue. They integrate these new data with existing datasets to define and characterize topologically associated domains (TADs) and chromatin loops. They benchmark their new data by comparing results between biological replicates and across different cell types. Then, they compare TAD boundaries with H3K27ac-defined super enhancers, gene expression, eQTL analyses, and GWAS loci. From their analyses, the authors conclude that TAD insulation is stronger if a super enhancer is located centrally rather than near the edge of that TAD and that functional links between variants and their corresponding genes are revealed by merging Hi-C with eQTL or GWAS analyses.

In this reviewer’s opinion, the major strength and noteworthy result of this work is the generation and quality assessment of the novel genome-wide HiC dataset. The authors’ integrative approach convincingly demonstrates the quality of these data and their potential for addressing the role of chromatin conformation in regulation of retinal gene expression. By making this resource available, the authors fill a major gap in the field, enhance the ability to pair regulatory elements with target genes, and advance efforts to interpret non-coding variants in the context of retinal disease. I would suggest the following recommendations to increase the impact of the work:

We sincerely appreciate the reviewer’s generous comments. The reviewer’s valuable and insightful suggestions have led to improvement of our manuscript. We do hope this manuscript will be a valuable resource for scientists working in diverse areas including gene regulation, human genetics and vision science.

Major suggestions:

1. I would encourage the authors to make this important resource as accessible as possible for other researchers, for example by providing “.hic” files for visualization of contacts in applications like Juicebox, the UCSC genome browser, or equivalent. Similarly, facilitating easy visualization of chromatin loops would allow for informal browsing of the data, which will likely allow more users to appreciate and build upon this important work. For the same reason, please provide a table of identified super enhancers in a “.bed” or equivalent format.

We thank the reviewer for the suggestion. In our GEO submission, we have provided .hic files to allow for direct exploration of the contact maps. We have also provided .bed files for mapping super enhancers, TADs, and accessible footprints as well as a .bedpe file for loops. These results should allow for other researchers to easily explore their regions of interest in depth. In addition,

we created a shiny application (grn.nei.nih.gov) to allow less-tech-savvy users investigate the genomic architecture in specific loci.

2. The authors should consider revising some statements regarding “functionality” since strictly speaking the analyses presented here are based on correlations and associations. The authors could strengthen the connection of this work to biological function by performing further experiments, but I believe that would be beyond the scope of this manuscript. I don’t think it would warrant delaying the publication of these data.

We have revised statements regarding functionality on lines 32, 278, 280, 396-397, 417 and 450. We agree that further experiments will delay the manuscript and prove detrimental to the value of this work. Several groups are waiting anxiously for the raw data to be released to perform follow up work on their preferred genes/loci.

3. I expect this work will be broadly read and cited, so I think a frank assessment of the strengths and weakness of HiC data in the Discussion is warranted. For example, there is a commonly held idea that chromatin conformation data can be used to unambiguously pair enhancers with their functional targets. Given the depth and resolution of their data, I would like the authors to comment on to what degree this is possible and what limitations remain. A very brief mention of the technical limitations and potential artifacts of HiC should also be included.

We thank the reviewer for the comment and suggestion. We included additional text about Hi-C and the resolution of our data in the discussion section, as follows (lines 325-335):

“Hi-C allows the exploration of the 3D genome architecture by identifying chromatin contacts but can only reveal pairwise interactions. Thus, it is impossible to determine if several regions interacting together in a Hi-C contact map coexist in each cell of the population or if they reflect a heterogeneous cell population each with unique pairs of interacting regions. At high resolution, fine regulatory structures can be resolved such as promoter-enhancers interactions. Our data have a resolution of almost 3kb, which means that significant chromatin contacts spanning over just few kilobases can be resolved. However, this average resolution is not homogeneous across the genome; some regions need higher sequencing depth to reach this resolution while some others exceed this resolution at our sequencing depth. This heterogeneity is considered by statistically calling significant contacts at each genomic location.”

4. Several figures in the paper make conclusions from quantitative data without reporting specific statistical analysis. The authors should perform and report the appropriate statistics where relevant. Such panels include 1E, 2C, 2D, 2E, 2F, 4A, 4C, 4D, 6A, 6B, 6E, and 6F.

To address this, we have provided additional statistical information for panels 1E (lines 110-112; 849-851), 4A (line 197-199), 4C (lines 202-204), 4D (lines 206-209), 6B (lines 924-926), and 6F (lines 273-275).

Figures 2D and 2F are intended to qualitatively compare the contact maps of different tissues and species but we do not provide a quantitative analysis. The data presented in Figures 2C, 2E, 6A, and 6E are not replicated therefore statistical testing is not possible and we have avoided making any claims about significance in the corresponding text.

Minor comments:

1. The authors use the NRL locus as an example in Figure panels 1D, 3F, and 6G. In each

case, it appears that a different isoform of NRL is shown. Please clarify which isoform is most representative for the adult human retina or why different isoforms are shown in each instance.

The primary isoform is ENST00000561028. Figure 1D shows this isoform although it is difficult to make out the individual exons in the original figure, we have modified this panel to make it clearer. Figure 3F shows this isoform and therefore we made no adjustments here. Figure 6G inadvertently showed the first exon as translated rather than as UTR; we have updated this panel to correct the error.

2. Please clarify in the online methods the number of retinas from female and male donors that were used for the HiC vs the CUT&RUN analyses.

We have now added the number of retinas from female and male donors that were used for the HiC (lines 473-474) and CUT&RUN analyses (line 486) in the revised manuscript.

3. There are a few grammatical and typographical errors throughout the manuscript that should be addressed, including:

- a. lines 68-71 is missing a preposition between “little” and “half”
- b. Panels 3B and 3C are switched in figure 3 relative to the text in lines 168-173
- c. Line 522 “Read mapping on the human genome” should be “Reads mapping on the human genome”.

We have now corrected the grammatical and typographical errors throughout the manuscript as suggested by the reviewer.

4. Lines 141-144 describing 2F are somewhat confusing and may benefit from being split into two sentences.

We have now modified the lines describing figure 2F (lines 142-145).

5. The authors should clarify why they are comparing raw contact counts in Figures 2D and 2F. Would this be expected to be biased according to sequencing depth differences between the datasets?

We have modified these two panels to show Knight-Ruiz normalized counts instead of raw counts for more accurate cross-experiment comparisons. We have updated the figure legend to reflect this change (lines 861, 867) in the revised manuscript.

REVIEWERS' COMMENTS

Reviewer #1 (Remarks to the Author):

The authors addressed the reviewer's concerns and comments. I agree this study will be a valuable resource for vision researchers with the new genome-wide HiC dataset.

Reviewer #2 (Remarks to the Author):

I appreciate the authors efforts to address my previous suggestions and comments. There are still a few issues that should be addressed from the initial review process.

Regarding Major Suggestion #2: There are still a few places in the manuscript that describe "function" in terms that are not supported by the data presented. (e.g. lines 136, 206, 207).

Regarding Major Suggestion #3: I think the field of retinal gene regulation would be better served by explicitly stating in their revised text that "One weakness of the HiC approach is that the physical interaction between candidate enhancers and promoters does not directly demonstrate a functional relationship between these elements with respect to gene regulation."

Regarding Minor Comment #1: I'd like to point out that the issue with NRL isoform inconsistencies across figures is still present. The isoform in Fig. 1D shows the gene PCK2 within an intron of NRL, whereas in Fig. 3F and Fig. 6G the two genes do not overlap. Please indicate which is correct and cite the study that demonstrates which isoform of NRL is dominant in the human retina.

Response to the Reviewers' Comments

Reviewer #1 (Remarks to the Author):

The authors addressed the reviewer's concerns and comments. I agree this study will be a valuable resource for vision researchers with the new genome-wide HiC dataset.

We are grateful for the reviewer's time and comment.

Reviewer #2 (Remarks to the Author):

I appreciate the authors efforts to address my previous suggestions and comments. There are still a few issues that should be addressed from the initial review process.

We thank the reviewer for the time spent on our manuscript and apologize for the few remaining issues that we have addressed below.

Regarding Major Suggestion #2: There are still a few places in the manuscript that describe "function" in terms that are not supported by the data presented. (e.g. lines 136, 206, 207).

We have modified the lines 136, 206 and 207 (137, 206 and 208 in the revised version).

Regarding Major Suggestion #3: I think the field of retinal gene regulation would be better served by explicitly stating in their revised text that "One weakness of the HiC approach is that the physical interaction between candidate enhancers and promoters does not directly demonstrate a functional relationship between these elements with respect to gene regulation."

We have added this sentence in the discussion, lines 448-452.

Regarding Minor Comment #1: I'd like to point out that the issue with NRL isoform inconsistencies across figures is still present. The isoform in Fig. 1D shows the gene PCK2 within an intron of NRL, whereas in Fig. 3F and Fig. 6G the two genes do not overlap. Please indicate which is correct and cite the study that demonstrates which isoform of NRL is dominant in the human retina.

We did not find a citation for a specific retinal isoform of NRL. However, our lab has sequenced more than 500 human retinal samples over the past decade (See Ratnapriya et al. *Nat Genet.* 2019, ref 38). We have pulled the transcript-level data from this vast dataset and plotted the expression of each NRL transcript (see below).

Transcript ENST00000397002 is clearly the dominant transcript in human retina therefore we have updated our figures to show the isoforms in all cases. Note that this differs from the isoform identified as the MANE select (ENST00000561028). For analyses using all expressed genes, we have used the MANE select list to ensure reproducibility.